# CASZ1 induces skeletal muscle and rhabdomyosarcoma differentiation through a feed-forward loop with MYOD and MYOG

Zhihui Liu[1,3 ✉], Xiyuan Zhang[1,3], Haiyan Lei[1], Norris Lam[1], Sakereh Carter[1], Oliver Yockey[1], Max Xu[1], Arnulfo Mendoza[1], Edjay R. Hernandez[1], Jun S. Wei [2], Javed Khan [2], Marielle E. Yohe[1], Jack F. Shern[1] & Carol J. Thiele[1 ✉]

Embryonal rhabdomyosarcoma (ERMS) is a childhood cancer that expresses myogenic master regulatory factor MYOD but fails to differentiate. Here, we show that the zinc finger transcription factor CASZ1 up-regulates MYOD signature genes and induces skeletal muscle differentiation in normal myoblasts and ERMS. The oncogenic activation of the RAS-MEK pathway suppresses CASZ1 expression in ERMS. ChIP-seq, ATAC-seq and RNA-seq experiments reveal that CASZ1 directly up-regulates skeletal muscle genes and represses non-muscle genes through affecting regional epigenetic modifications, chromatin accessibility and super-enhancer establishment. Next generation sequencing of primary RMS tumors identified a single nucleotide variant in the *CASZ1* coding region that potentially contributes to ERMS tumorigenesis. Taken together, loss of CASZ1 activity, due to RAS-MEK signaling or genetic alteration, impairs ERMS differentiation, contributing to RMS tumorigenesis.

[1] Pediatric Oncology Branch, Center for Cancer Research, National Cancer Institute, Bethesda, MD, USA. [2] Genetics Branch, Center for Cancer Research, National Cancer Institute, Bethesda, MD, USA. [3] These authors contributed equally: Zhihui Liu, Xiyuan Zhang. ✉email: liuzhihu@mail.nih.gov; thielec@mail.nih.gov

Rhabdomyosarcoma (RMS) is the most common soft tissue sarcoma of childhood and an embryonal tumor morphologically resembling the developing skeletal muscle[1–7]. Multiple in vitro and in vivo models demonstrate that RMS arise from muscle stem cells, activated satellite cells and myoblasts[2–4,8], indicating that skeletal muscle progenitors are major contributors to RMS. Recent studies show that RMS can also arise from adipocytes and epithelial progenitor cells[6,7]. There are two major subtypes of RMS, *PAX-FOXO1* fusion-negative embryonal RMS (ERMS) and *PAX-FOXO1* fusion-positive alveolar RMS (ARMS)[9]. The RAS-RAF-MEK-ERK MAP kinase (RAS-MEK) pathway is often mutationally activated in ERMS contributing to a myogenic differentiation block, which can be ameliorated pharmacologically via MEK inhibition[4,10]. Myogenic regulatory factors (MRFs) MYF5, MYOD, MYOG and MRF4 are transcription factors that are essential for skeletal myogenesis[11]. It is not clear why MYOD fails to appropriately regulate expression of target genes and induce terminal differentiation despite adequate expression in RMS[1,12].

CASZ1, a zinc finger transcription factor, is highly expressed in embryonic somites and skeletal muscle of *xenopus*, embryonic somites of mouse and skeletal muscle of the adult human[13–15]. During development, CASZ1 homologs orchestrate cell-fate specification, commitment and differentiation in neuroblasts, retinal progenitors, T helper cells, cardiac progenitors, and cardiomyocytes[14,16–23]. The *CASZ1* gene encodes two isoforms. Human CASZ1a, has 1759 amino acids (AA) with 11 TFIIIA class C2H2 zinc fingers, while CASZ1b is the more evolutionarily conserved isoform and comprises the first 1166 AA of CASZ1a, but lacks six zinc fingers at the C-terminus[15]. Both isoforms function similarly to suppress neuroblastoma growth and regulate expression of neuronal genes[24–26]. However, they have been shown to play distinct roles in murine retina progenitor cells[16]. Allelic loss of *CASZ1* and epigenetic suppression of the remaining allele has been implicated in neuroblastoma tumorigenesis[24,25,27–29]. However, the contribution of CASZ1 to skeletal myogenesis and differentiation remains a gap in our knowledge. Moreover, whether CASZ1 contributes to RMS tumorigenesis or plays a role in its differentiation block has not been examined.

In this study, we find that CASZ1 expression is negatively regulated by aberrant RAS-MEK signaling that is a characteristic of ERMS. CASZ1 also plays a critical role in inducing skeletal myogenesis and co-operating to form a feed-forward loop with MYOD and MYOG that is critical for ERMS differentiation.

## Results

**CASZ1 is directly regulated by MRFs in myoblasts.** Consistent with previous studies, the expression of Casz1 was observed in the somites of the E12.5 mouse embryos, from which skeletal muscle precursors originate (Supplementary Fig. 1a). Knockout of Casz1 in mice is embryonic lethal[14]. Thus, we used mouse C2C12 myoblasts, a well-characterized in vitro skeletal muscle differentiation model to study the regulation and function of CASZ1 during myogenesis. Both isoforms of the *Casz1* gene were upregulated when C2C12 cells were cultured in differentiation medium (DM) compared to growth medium (GM) (Fig. 1a, left panel). Consistently, *Casz1* messenger RNA (mRNA) levels increased when C2C12 cells were induced to differentiate (Fig. 1a, right panel). Interrogation of published chromatin immunoprecipitation followed by DNA sequencing (ChIP-seq) data indicated that in C2C12 myotubes (MT) both MyoD and Myog bind to the promoter and enhancer regions of *Casz1* (Fig. 1b and Supplementary Data 1). Silencing of either MyoD or Myog but not Myf5 in C2C12 cells decreased Casz1 expression (Fig. 1c, d), which indicates that MyoD and Myog directly regulate *Casz1* transcription.

**CASZ1 regulates normal skeletal myogenesis.** We next evaluated whether CASZ1 plays a role in regulating myogenesis. Knockdown of Casz1 using either small-interfering RNA (siRNA) or short-hairpin RNA (shRNA) in differentiating C2C12 cells significantly decreased mRNA levels of skeletal muscle differentiation markers alpha actin (*Acta1*) and creatine kinase (*Ckm*), and protein levels of myosin heavy chain (MHC) (Fig. 1e, f). Moreover, loss of Casz1 attenuated myotube formation and decreased both the differentiation index and fusion index in C2C12 cells cultured in DM (Supplementary Fig. 1b, c left panel and Fig. 1g). When either the human CASZ1a or CASZ1b isoform were over-expressed, they upregulated skeletal muscle differentiation markers *Acta1*, *Ckm*, and MHC (Fig. 1h), and accelerated myotube formation shown by increased differentiation and fusion indices (Fig. 1i and Supplementary Fig. 1c, right panel).

**CASZ1 regulates MRFs and normal myogenesis.** To determine mechanisms by which Casz1 regulates myogenesis, we first investigated whether Casz1 regulates the expression of MRFs. Silencing of Casz1 increased Myf5 mRNA levels, but decreased MyoD and Myog mRNA levels (Fig. 1j). Conversely, over-expression of either CASZ1a or CASZ1b decreased Myf5 mRNA levels and increased MyoD and Myog mRNA levels (Supplementary Fig. 1d). Real-time PCR (RT-PCR) using a mouse Casz1-specific primer demonstrated that overexpression of CASZ1 upregulated endogenous *Casz1* gene expression (Supplementary Fig. 1d). Gene set enrichment analysis (GSEA) of RNA-seq data (Supplementary Data 2) demonstrated that loss of Casz1 led to a negative enrichment of MyoD signature genes and myogenesis genes (Fig. 1k). Consistently, RNA-seq data (Supplementary Data 2) from either CASZ1a or CASZ1b overexpressing C2C12 cells (GM 48 h) showed a positive enrichment of MyoD signature genes and myogenesis genes (Fig. 1l, m and Supplementary Fig. 1e). To further evaluate whether there are any functional differences between the two CASZ1 isoforms, we analyzed RNA-seq data when either isoform was overexpressed in C2C12 cells. A comparison of genes differentially regulated by either CASZ1a or CASZ1b (Supplementary Data 3) indicated that the changes in gene expression regulated by each isoform are positively correlated ($R^2 = 0.79$) (Supplementary Fig. 1f). Taken together, these findings indicate that CASZ1 regulates skeletal myogenesis by activating a feed-forward circuit to regulate expression of itself, MyoD, Myog, and skeletal muscle genes.

**CASZ1 is suppressed by activated RAS-MEK signaling in ERMS.** Having demonstrated that CASZ1 plays a role in myoblast differentiation, we hypothesized that CASZ1 may be dysregulated in RMS. Cell-fate and cell identity are determined by super-enhancers (SEs) that marked by extensive stretches of H3K27ac, and SEs are frequently dysregulated in cancers[30]; thus, we first queried public H3K27ac ChIP-seq data to examine the *CASZ1* gene locus during human skeletal myogenesis. We found a SE on the *CASZ1* gene locus that was present in human skeletal muscle myotubes but not in myoblasts (Supplementary Fig. 2a), which suggests that the increased expression of *CASZ1* is driven by a SE during human skeletal myogenesis. In ERMS, *CASZ1* expression is lower when compared to normal muscle in both the microarray data[31] deposited in R2 database and RNA-seq data acquired from the Integrated Rhabdomyosarcoma Databases (Supplementary Fig. 2b, c). We further evaluated mRNA levels of *CASZ1* by RT-PCR in normal skeletal muscle samples and RMS tumor samples. Consistently, we found that *CASZ1* levels are significantly lower in ERMS compared to normal skeletal muscle (Supplementary Fig. 2d). Moreover, we found that CASZ1

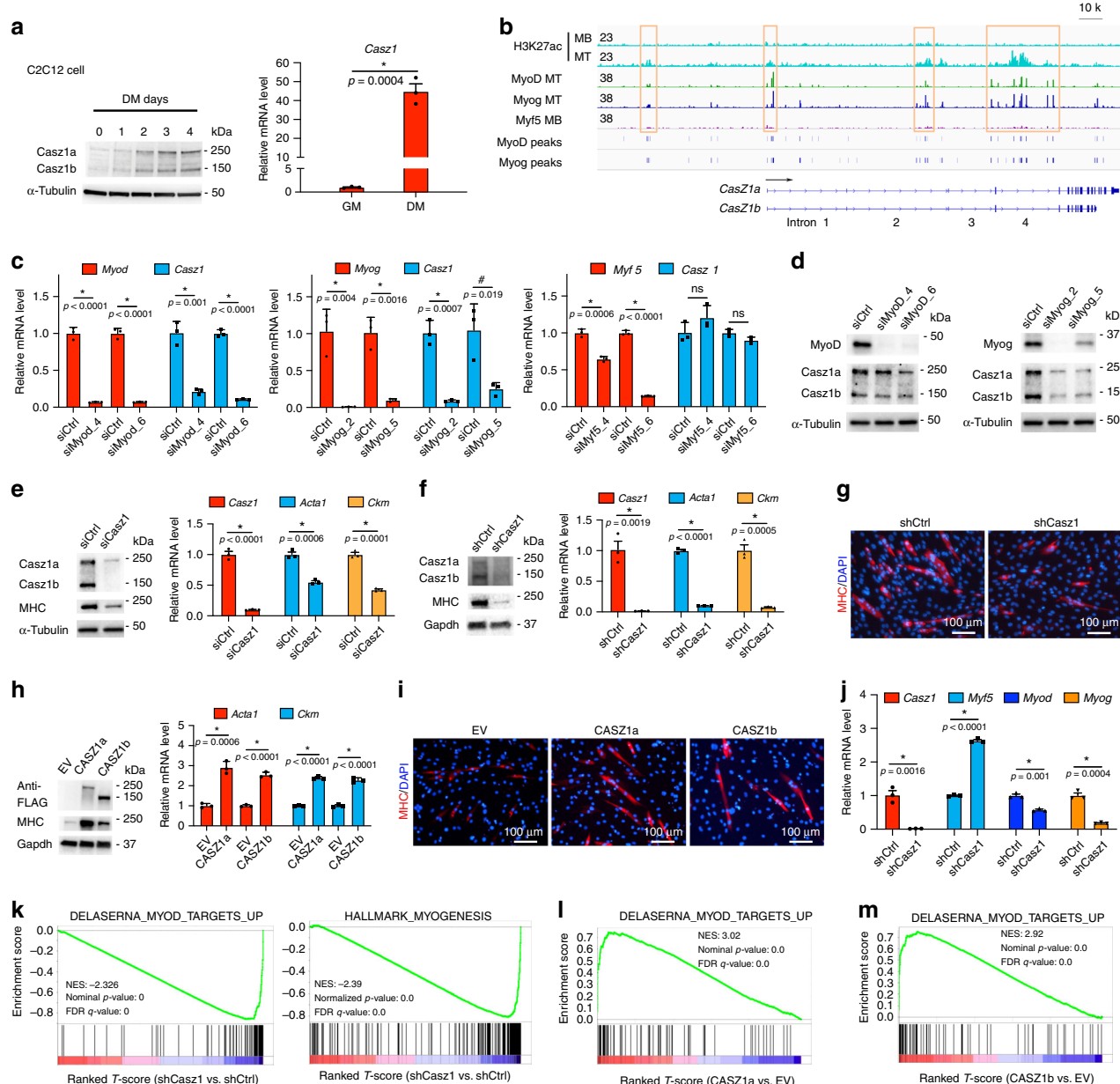

**Fig. 1 CASZ1 regulates skeletal myogenesis through cross-talking with MRFs. a** Both Casz1a and Casz1b protein levels and *Casz1* mRNA levels increase when C2C12 myoblasts are cultured in differentiation medium (DM) compared to growth medium (GM). **b** ChIP-seq of H3K27ac, MyoD, and Myog show their presence within *Casz1* genomic locus in C2C12 myotubes (MT) (representative peaks are highlighted by orange boxes); H3K27ac and Myf5 do not show significant binding within *Casz1* genomic locus in myoblasts (MB). The MyoD and Myog-binding peaks passed peak calling threshold of $p < 10^{-7}$ are shown under the signal tracks. **c** Knockdown of MyoD and Myog, but not Myf5 using siRNAs for 24 h in DM (total 48 h) significantly decreases *Casz1* mRNA levels compared to siRNA control (siCtrl). **d** Knockdown of MyoD and Myog decreases both Casz1a and Casz1b protein levels. **e** and **f** Knockdown of Casz1 in C2C12 cells using siRNA or shRNA attenuates the induction of MHC protein, *Acta1* and *Ckm* mRNA (DM 24 h). **g** Knockdown of Casz1 in C2C12 cells disrupts myotube formation (DM 3 days) shown by MHC (red) and DAPI (blue) staining. **h** Over-expression of either human CASZ1a or CASZ1b in C2C12 cells increases MHC protein levels and *Acta1* and *Ckm* mRNA levels (DM for 24 h, total 48 h). **i** Overexpression of either human CASZ1a or CASZ1b in C2C12 cells (DM 2 days) accelerates myotube formation as shown by MHC (red) or DAPI (blue) staining. **j** Knockdown of Casz1 in C2C12 cells (DM 24 h) increases *Myf5* mRNA levels and decreases *MyoD* and *Myog* mRNA levels. **k** GSEA shows negative enrichment of MyoD signature genes and myogenesis genes after knockdown of Casz1 in C2C12 cells (DM 2 days). **l** and **m** GSEA shows the positive enrichment of MyoD signature genes after over-expression of either human CASZ1a or CASZ1b in C2C12 cells (GM 2 days). Data represent mean ± SEM, *n* = 3 biological replicates, ns not significant. Two-sided Student's *t*-test was used to calculate statistical difference. Source data are provided as a Source Data file.

expression was significantly upregulated in the ERMS cell line, RD, when cultured in DM (Supplementary Fig. 2e).

The RAS-MEK pathway is frequently mutationally activated in ERMS patients[4,9,10]. Treatment of the RAS mutated ERMS cell lines RD and SMS-CTR with trametinib, an inhibitor of MEK1/2 (MEKi) induces skeletal muscle differentiation[10]. We observed

that MEKi treatment of ERMS resulted in an increase in CASZ1 expression (Fig. 2a and Supplementary Fig. 2f). Similar to pharmacologic inhibition, genetic knockdown of MEK1 in RD cells led to a significant increase in *CASZ1* mRNA levels (Supplementary Fig. 2g). Consistently, we observed the acquisition of a SE at the *CASZ1* gene locus in MEKi induced

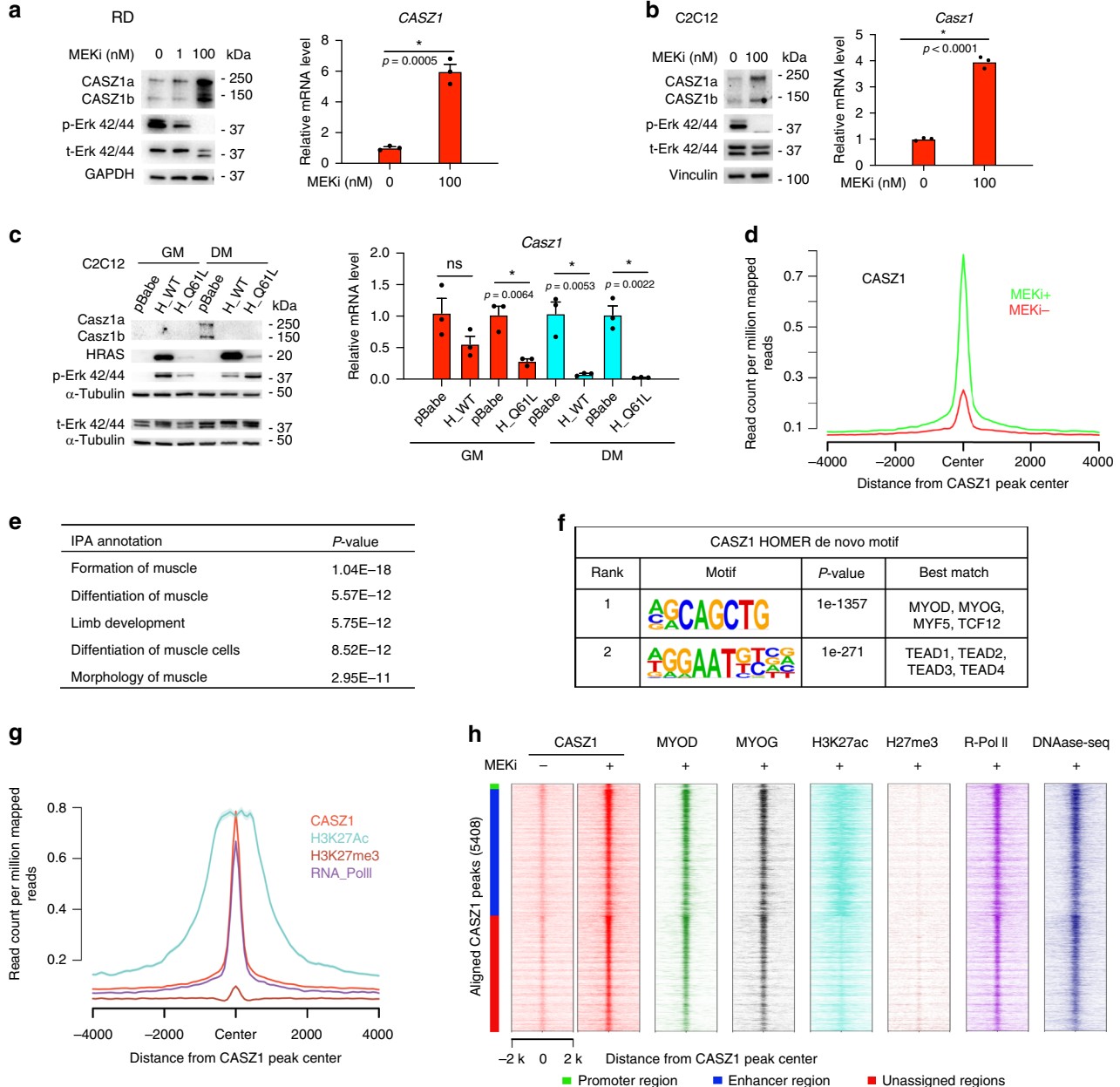

**Fig. 2 Global binding of CASZ1 on genomic DNA in differentiated SMS-CTR cells. a** MEK inhibitor trametinib (MEKi) treatment of RD cells leads to a decrease of phosphor-Erk 42/44 and an increase of protein levels of CASZ1a and CASZ1b (left panel), as well as an increase of mRNA levels of *CASZ1* (right panel). **b** MEKi treatment of C2C12 cells results in a decrease of phospho-Erk42/44 and an increase of *CASZ1* mRNA levels. **c** Overexpression of either wild-type HRAS (H_WT) or mutant HRAS (H_Q61L) in C2C12 cells leads to an increase of phospho-Erk 42/44 and a decrease of protein levels of CASZ1a and CASZ1b (left panel), as well as a decrease of mRNA levels of *CASZ1* (right panel). **d** ChIP-seq was performed using anti-CASZ1 antibody in SMS-CTR cells with or without MEKi treatment. Composite plot shows the increase of endogenous CASZ1 binding on genomic DNA after SMS-CTR cells are treated with 100 nM MEKi for 48 h. **e** Ingenuity pathway analysis (IPA) shows that CASZ1-binding site-associated genes are enriched in muscle formation and differentiation. **f** Homer de novo motif scan shows that CASZ1-binding peaks are enriched in motifs similar to MRFs-binding motifs and TEAD-binding motifs. **g** Composite plot shows that CASZ1-binding peaks are co-occupied by H3K27ac, RNA Pol II but not H3K27me3. **h** ChIP-seq heatmap shows the top ranked CASZ1-binding peaks (5408) and the aligned peaks of MYOD, MYOG, H3K27ac, RNA Pol II with DNase I-seq at CASZ1 peak center. Among 5408 CASZ1-binding sites, 125 binding sites are within promoter region (green bar), 2771 binding sites are within enhancer region (blue bar), and 2512 binding sites are within undefined region (red bar). Data represent mean ± SEM, $n = 3$ biological replicates. Two-sided Student's *t*-test was used to calculate statistical difference. Source data are provided as a Source Data file.

differentiated SMS-CTR cells (Supplementary Fig. 2h) in H3K27ac ChIP-seq data[10]. To investigate whether the RAS-MEK pathway regulates CASZ1 in myoblasts, we performed a loss of function study in C2C12 cells and found that either genetic knockdown of Mek1 or pharmaceutical inhibition of MEK using trametinib resulted in an increase of Casz1 expression (Fig. 2b

and Supplementary Fig. 2i). Overexpression of wild-type HRAS or patient-derived HRAS_Q61L mutant in C2C12 cells activated the RAS-MEK pathway led to a decrease of Casz1 expression (Fig. 2c). Overexpression of other activating RAS mutation constructs KRAS_G12D and NRAS_Q61K in C2C12 cells also caused a significant decrease in Casz1 mRNA levels

(Supplementary Fig. 2j). Moreover, MEKi treatment of a KRAS mutated neuroblastoma cell line NBEB (KRAS_G12D) also increased CASZ1 expression (Supplementary Fig. 2k). These results demonstrate that CASZ1 is regulated by activation of RAS-MEK signaling in normal skeletal muscle cells and in two different types of tumor cells.

**CASZ1 binds to the regulatory regions of muscle genes**. The global-binding sites of CASZ1 on genomic DNA have not been identified to date. To gain insights into the mechanisms by which CASZ1 regulates gene transcription to influence cell-fate we performed ChIP-seq. Remarkably, 5408 CASZ1-binding sites accompanied by increased signal intensity were found in differentiated SMS-CTR cells while only 64 binding sites were found in undifferentiated cells (Supplementary Data 4 and Fig. 2d). Peak distribution analysis showed that in the differentiated SMS-CTR cells, 2% of CASZ1-binding sites are at promoter regions (defined by + 1 kb of transcription start site (TSS)) and over 90% of CASZ1-binding sites are within intronic or intergenic regions of MEKi treated SMS-CTR cells. Ingenuity pathway analysis (IPA) of CASZ1-binding site-associated genes ($n = 2916$) identified significant enrichment in genes involved in skeletal and muscular system development and function (Supplementary Fig. 2m and Fig. 2e). Homer de novo motif scan of the discovered CASZ1-binding sites identified E-box as the 1st highest-ranked motif, and this motif is best matched to the binding motif of MRFs and TCF12 (Fig. 2f). The 2nd highest-ranked motif was TEAD family protein-binding motifs. TEAD family members are required for muscle differentiation[32]. To gain further insight into the epigenetic modifications associated with CASZ1-binding sites, we investigated the chromatin landscape in SMS-CTR cells that have been treated with MEKi using previously published data (GSE85169, GSE85171). Composite plot and ChIP-seq heatmaps showed that CASZ1 peak centers overlapped with H3K27ac, RNA Pol II-binding sites and DNase I hypersensitivity regions, but not repressive histone mark H3K27me3-binding sites (Fig. 2g, h). Overlaying the enhancers that were identified using H3K27ac ChIP-seq results with CASZ1-binding sites in trametinib treated SMS-CTR cells, we found that 2771 CASZ1-binding sites (51.24%) are within enhancers (Fig. 2h).

**CASZ1 is a component of a CRC in differentiated ERMS**. By comparing CASZ1 and MYOD or MYOG ChIP-seq datasets generated from MEKi treated SMS-CTR cells, we found genome-wide co-occupancy among these transcription factors (Fig. 2h). Among the ~5400 CASZ1 peaks observed post-MEKi treatment (day 2), ~4100 loci (76%) were also bound by MYOD, ~2400 loci (45%) were also bound by MYOG; and ~2350 loci (44%) were bound by all three (Fig. 3a and Supplementary Data 5). IPA revealed that genes bound by both CASZ1 and MYOD or MYOD alone were predominantly involved in skeletal and muscular system development and function (Supplementary Fig. 3a, b), while genes bound by CASZ1 alone were predominantly involved in nervous system development and function (Supplementary Fig. 3c). A recent study identified a core transcriptional regulatory circuitry (CRC) composed of MYOG, MYOD and a few other transcription factors in MEKi-induced differentiated SMS-CTR cells[10]. A CRC comprises a group of transcription factors (TFs), marked by the presence of super-enhancers (SEs). These CRC TFs not only bind to their own gene loci and regulate their own gene expression, but also mutually regulate each other member in the CRC, thus forming an interconnected auto-regulatory feed-forward loop[33]. CASZ1 is a component of the CRC together with MYOD and MYOG in differentiated SMS-CTR cells, based on (1) CASZ1 expression is regulated by itself (Supplementary Fig. 1d)

and is driven by a SE (Supplementary Fig. 2a, h); (2) CASZ1, MYOD and MYOG bind consensus DNA sequences adjacent to each other within their own SE regions (Fig. 3b–d); (3) CASZ1, MYOD and MYOG positively regulate each other's expression in muscle cells (Fig. 1c, j and Supplementary Fig. 1d); and (4) among ~700 SEs in the differentiated SMS-CTR cells, over 80% contain CASZ1, MYOD, and MYOG-binding motifs based on motif scan using the CRC analysis tool[33]. These findings indicate that after MEKi treatment, these transcription factors become the components of a CRC that form a feed-forward autoregulatory loop to induce skeletal muscle differentiation (Fig. 3e). Among the SE-driven genes that were co-occupied by CASZ1, MYOD, and MYOG, there were multiple skeletal muscle differentiation markers, such as TNNC2 and TNNI1 (Supplementary Fig. 3d). In addition, MIR206, a key microRNA that regulates skeletal myogenesis and functions as a tumor suppressor in RMS[34], was also found to be driven by a SE co-occupied by CASZ1, MYOD and MYOG (Supplementary Fig. 3e). MEKi treatment upregulates CASZ1 expression and leads to CASZ1 occupancy at genes involved in skeletal myogenesis, suggesting that CASZ1 contributes to MEKi induced myogenic program. Consistently, we found that the knockdown of CASZ1 in SMS-CTR cells attenuated MEKi induced upregulation of the protein levels of MHC, and the mRNA levels of skeletal muscle differentiation markers TNNC2, TNNI2, and MYOG (Fig. 3f, g).

**CASZ1b regulates RMS cell differentiation**. To determine whether CASZ1 directly regulates ERMS differentiation as observed in normal myoblasts (Fig. 1), we utilized the more evolutionarily conserved CASZ1b isoform[24,26] to generate a doxycycline (Dox) inducible, 3xTy1-tagged, CASZ1b expressing SMS-CTR cell line (CTRtetCASZ1b) and a tetracycline (Tet) inducible, FLAG tagged, CASZ1b expressing RD cells line (RDtetCASZ1b) (Fig. 4a). Cell confluence assays showed that restoration of CASZ1b suppressed cell proliferation in both RD cells and SMS-CTR cells (Fig. 4b). CASZ1b upregulates the protein levels of the skeletal differentiation markers, MHC and MYOG in RD cells (Fig. 4c), despite the obeservation of only a few multinucleated cells in RD cells when CASZ1b was restored. In SMS-CTR cells, phalloidin and DAPI staining showed the formation of multinucleated cells (> 30%) reminiscent of the terminal differentiation of muscle cells upon restoration of CASZ1b in SMS-CTR cells (Fig. 4d). To investigate whether CASZ1 is relevant in ARMS, we generated a Tet inducible CASZ1b-overexpressing stable clone using RH30 ARMS cell line. We found that over-expression of CASZ1b in RH30 cells resulted in a decrease of cell proliferation and an upregulation of skeletal muscle differentiation markers MHC, ACTA1, and CKM, as well as the myogenic regulatory factors MYOD and MYOG (Supplementary Fig. 4a–c). These results indicate that CASZ1 induces muscle differentiation and suppresses tumor cell proliferation in the ARMS RMS subtype as well.

To further interrogate the transcriptional consequences of CASZ1 expression in ERMS, we performed RNA-seq analysis (gene list is shown in Supplementary Data 6). A customized human myogenic differentiation signature (HSMM_UP)[10] was integrated into GSEA. We found that the restoration of CASZ1b in either SMS-CTR or RD cells leads to a positive enrichment of a MYOD and a human skeletal muscle differentiation signature. In contrast, E2F target genes and pRb repressed genes were negatively enriched (Fig. 4e–h, Supplementary Fig. 4d, and Supplementary Data 7). Consistent with the RNA-seq data, RT-PCR and western blot analyses showed that restoration of CASZ1b in RD cells led to the down regulation of cell-cycle genes CCNB1, CCNA2, CDK1 and PLK1 (Supplementary Fig. 4e, f).

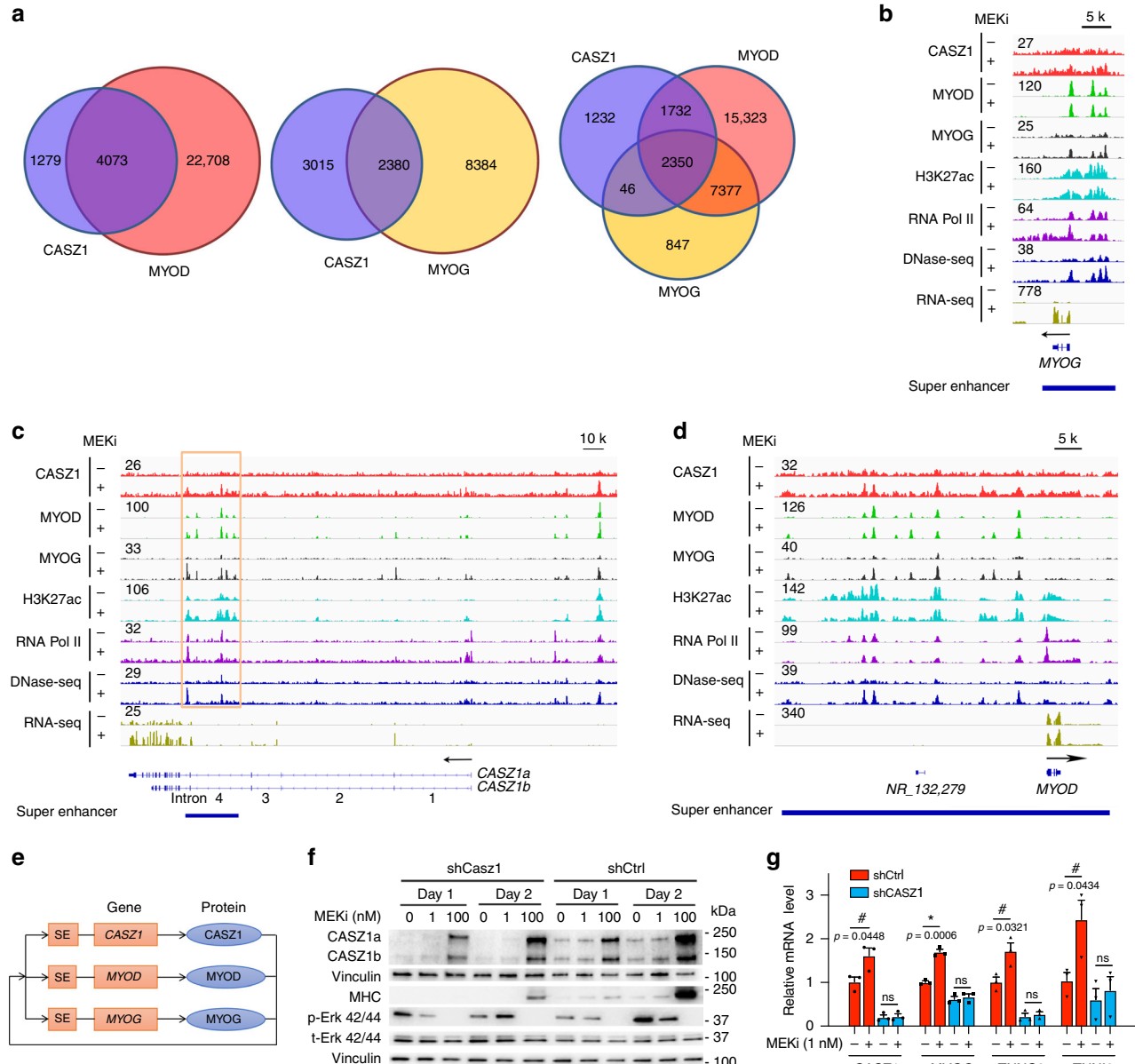

**Fig. 3 CASZ1 is a component of a CRC in MEKi induced, differentiated ERMS. a** Venn-diagram shows the overlap between CASZ1, MYOD, and MYOG-binding sites in MEKi induced, differentiated SMS-CTR cells. **b–d** Signal tracks show the binding and co-occupancy of CASZ1, MYOD, MYOG, H3K27ac, RNA Pol II on CASZ1, MYOD, and MYOG gene locus in SMS-CTR cells before (–) and after (+) MEKi induced differentiation. DNase I-seq and RNA-seq signals on these genes are also shown. **e** A graphic shows the CRC components of CASZ1, MYOD and MYOG in MEKi-induced, differentiated SMS-CTR cells. Orange square box marked with SE represents a super-enhancer; orange rectangular box marked with a gene name represents a gene; blue rectangular oval box marked with a gene name represents a protein. **f** Knockdown of CASZ1 in SMS-CTR cells attenuates the ability of MEKi to induce the skeletal muscle differentiation marker MHC as detected by western blots. **g** One nanomolar MEKi treatment of SMS-CTR cells for 3 days significantly increases skeletal muscle genes *MYOG, TNNC2,* and *TNNI2* mRNA levels compared to controls, but knockdown of CASZ1 in SMS-CTR cells attenuates MEKi induced upregulation of these genes. Data represent mean ± SEM, *n* = 3 biological replicates, ns not significant. Two-sided Student's *t*-test was used to calculate statistical difference. Source data are provided as a Source Data file.

To compare programs regulated by CASZ1 induced myoblast differentiation to those regulated during CASZ1 induced ERMS differentiation, we analyzed differentially expressed genes (RNA-seq) by utilizing GSEA and the C2-curated gene sets. In both models, CASZ1b induced genes are enriched in MYOD targets while CASZ1 repressed genes are enriched in EGFR signaling and RB1 repressed gene signature (Supplementary Data 8). However, in the normal cell model (C2C12), CASZ1 regulates gene sets associated with repression of oligodendrocyte differentiation, telomere maintenance and chromosome maintenance, while in

the tumor model (SMS-CTR cells) the top regulated gene sets are involved in breast cancer development, colorectal cancer development, cervical cancer proliferation (Supplementary Data 8). Taken together, CASZ1 induces skeletal muscle differentiation program in both myoblasts and ERMS, however, CASZ1 preferably regulates normal development processes in myoblasts and genes associated with cancer progression in ERMS.

To investigate whether CASZ1 interacts with MYOD and MYOG to regulate the expression of muscle genes, we performed co-immunoprecipitation in CASZ1b restored SMS-CTR cells

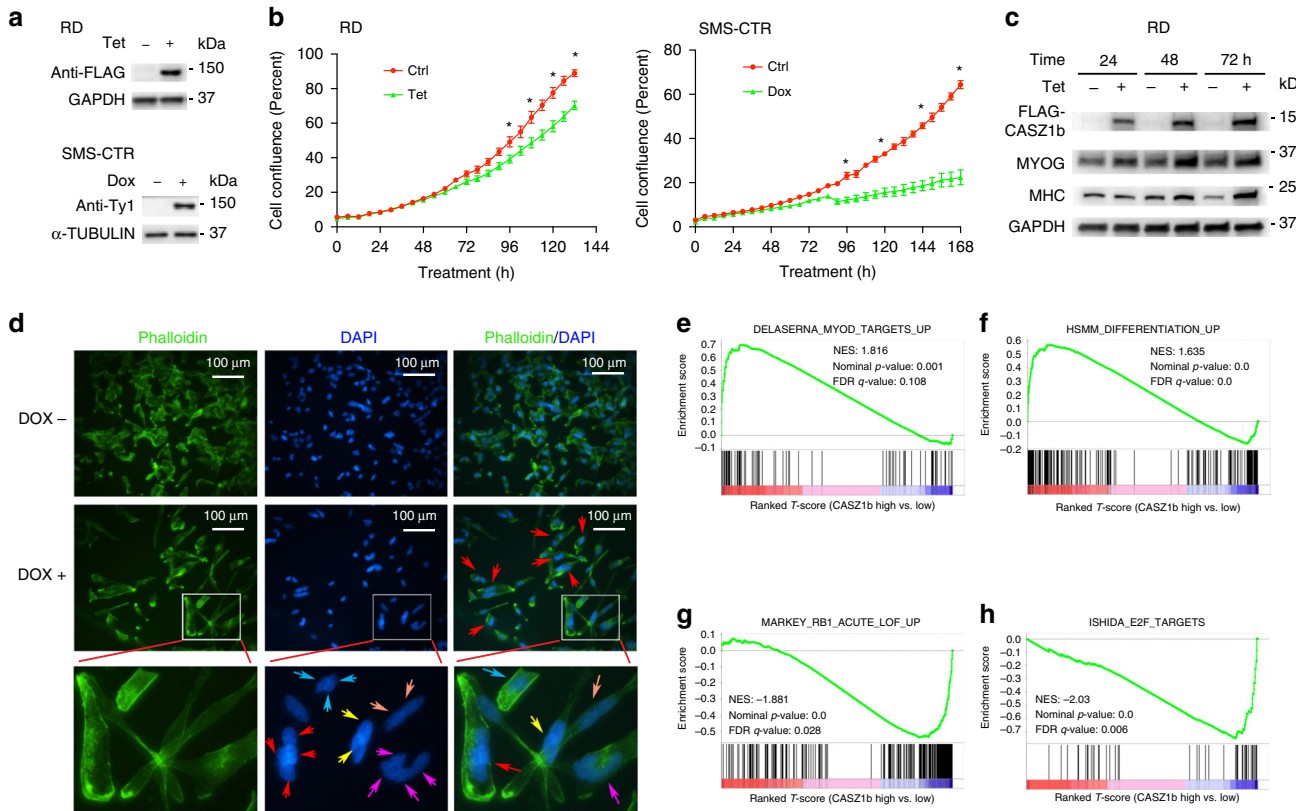

**Fig. 4 Restoration of CASZ1b in ERMS induces cell differentiation. a** FLAG tagged or 3xTy1 tagged CASZ1b is stably cloned into RD and SMS-CTR cells. CASZ1b expression is Tetracycline (Tet) or Doxycycline (Dox) inducible shown by western blot. **b** Restoration of CASZ1b in RD cells (Tet treatment) or SMS-CTR cells (Dox treatment) inhibits cell proliferation compared to control cells (Ctrl) shown by IncuCyte cell confluence assay. **c** Restoration of CASZ1b in RD cells increases expression of skeletal muscle differentiation markers MYOG and MHC protein shown by western blot. **d** Staining of SMS-CTRtetCASZ1b cells with Phalloidin (red) and DAPI (blue) shows that cells with restoration of CASZ1b (Dox +), but not control (Dox-) form multinuclear cells (middle panel, red arrow). Bottom panel shows the enlarged image from the middle panel, and the arrow with the same color indicates these nuclei are in one cell. **e**–**h** GSEA assays of RNA-seq data show the positive enrichment of MYOD signature genes and human skeletal muscle differentiation genes after the restoration of CASZ1b in SMS-CTR cells, and negative enrichment of Rb repressed genes and E2F target genes. Data represent mean ± SEM for technical triplicates, *$p < 0.01$. Two-sided Student's $t$-test was used to calculate statistical difference. Source data are provided as a Source Data file.

using an anti-CASZ1 antibody. There was no evidence of an interaction of CASZ1b with MYOD or MYOG (Supplementary Fig. 4g). Taken together, our results show that mechanistically CASZ1b repressed E2F target genes and cell cycle genes to suppress ERMS cell proliferation, while upregulating MYOD signature genes and skeletal muscle differentiation genes to induce ERMS differentiation.

**CASZ1b directly regulates muscle and neuronal genes in ERMS**. To investigate how CASZ1b regulates the chromatin landscape to control gene transcription and induce ERMS differentiation, we performed ChIP-seq in SMS-CTRtetCASZ1b cells in the presence and absence of Dox for 2 days, using both anti-Ty1 and anti-CASZ1 antibodies, and we found a significant correlation in the observed CASZ1b-binding peaks (Supplementary Fig. 5a and Supplementary Data 9). Using the CASZ1 antibody, we identified 15,772 CASZ1-binding sites that are associated with 6271 genes in Dox-treated cells compared to 128 binding sites in control cells. The increase in CASZ1 signal intensity was shown in Supplementary Fig. 5b. The top ranked motif of CASZ1b-binding sites is best matched to the binding motif of AP-1 subunits and the binding motifs of MRFs (Supplementary Fig. 5c). Similar to the endogenous CASZ1-binding sites observed in the MEKi-induced differentiated SMS-CTR cells (Fig. 2f), E-box is a top ranked motif identified in both

conditions. Consistent with the results from MEKi treated SMS-CTR cells, the CASZ1b bound peak center overlaps with active histone marks H3K27ac and H3K4me3, RNA Pol II-binding sites, but not the repressive histone mark H3K27me3 (Supplementary Fig. 5d–f). The chromatin accessibility was further delineated using ATAC-seq, which indicated that the CASZ1b peak center is co-occupied with open chromatin. When focused on all these ~15,800 CASZ1b-binding peaks, no obvious H3K27ac, H3K4m3, RNA Pol II or ATAC-seq signal changes were observed before or after the restoration of CASZ1b in SMS-CTR cells (Supplementary Fig. 5f).

To understand the regulatory influence of CASZ1b binding, we focused on genes both bound and transcriptionally regulated by CASZ1b. We compared RNA-seq data and CASZ1b ChIP-seq data from SMS-CTRtetCASZ1b cells treated with or without Dox for 2 days. In this analysis, 586 genes were found to be bound and transcriptionally regulated by CASZ1b (Supplementary Data 10). IPA showed that genes directly upregulated by CASZ1b are involved in skeletal and muscular system development and some other pathways (Fig. 5a and Supplementary Fig. 5g); and genes directly downregulated by CASZ1b are involved in nervous system development and some other pathways (Fig. 5a and Supplementary Fig. 5h). The skeletal muscle genes directly upregulated by CASZ1 include *MYOD*, *MEF2D*, *SIX1*, and *TNNT2* (Fig. 5b, left panel). The skeletal muscle system genes that are CASZ1b bound and transcriptionally downregulated are

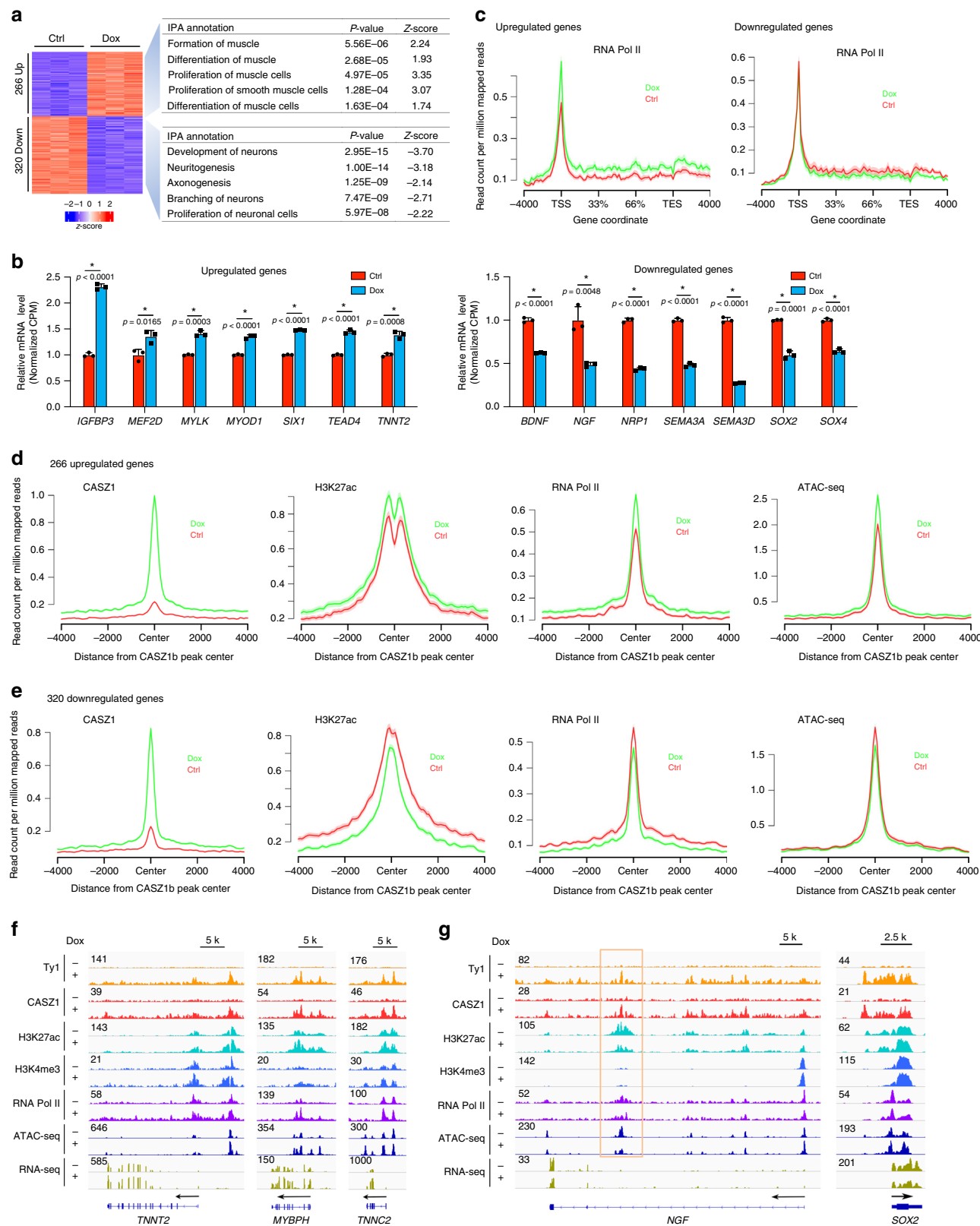

enriched in genes involved in muscle cell movement (Supplementary Fig. 5i), which is a critical step involved in myoblast fusion[35]. The neural genes directly downregulated by CASZ1 include *BDNF*, *NGF*, *NRP1*, and *SOX4* (Fig. 5b, right panel). To investigate how many skeletal muscle differentiation genes were directly upregulated by CASZ1b, we focused on the 98

upregulated genes derived from the HSMM_DIFFERENTIA-TION_UP and MYOD_TARGETS _UP gene sets when CASZ1b was restored in SMS-CTR cells (Fig. 4e, f and Supplementary Data 7). CASZ1b ChIP-seq results indicate that almost 40% (40/90 genes) of the directly upregulated skeletal muscle differentiation genes have an average of three CASZ1b-binding sites per

**Fig. 5 CASZ1b directly upregulates skeletal muscle genes and downregulates neuronal genes. a** RNA-seq heatmap shows genes transcriptionally regulated and bound by CASZ1 based on RNA-seq data and ChIP-seq data (left panel) from SMS-CTRtetCASZ1b cells treated with or without Dox. Ingenuity pathway analysis (IPA) shows that CASZ1b directly upregulated genes are enriched in muscle formation and differentiation, but CASZ1b directly downregulated genes are enriched in development of neurons. **b** Examples of skeletal muscle genes positively regulated by CASZ1b and neuronal genes negatively regulated by CASZ1b based on normalized RNA-seq reads (CPM counts per million). **c** Composite plot shows an increased signal of RNA Pol II on the gene body of upregulated genes and a slightly decreased signal of RNA Pol II on the gene body of downregulated genes. **d** and **e** Composite plot shows an increased signal of H3K27ac, RNA Pol II, and ATAC-seq at CASZ1b peak center of 266 upregulated genes and a slightly decreased signal of H3K27ac, RNA Pol II, and ATAC-seq at CASZ1b peak center of 320 downregulated genes. **f** Signal tracks show the increase of CASZ1b, H3K27ac, RNA Pol II binding and signals of ATAC-seq and RNA-seq on directly upregulated skeletal muscle genes *TNNT2*, *MYBPH*, and *TNNC2* before (-) and after ( + ) Dox treatment. **g** Signal tracks on downregulated neural genes *NGF* and *SOX2*. Decreased signal of H3K27ac and ATAC-seq was observed within *NGF* gene (orange box) but not *SOX2* gene before (-) and after ( + ) Dox treatment. Data represent mean ± SEM, $n = 3$ biological replicates. Two-sided Student's *t*-test was used to calculate statistical difference. Source data are provided as a Source Data file.

gene (Supplementary Data 11). The remaining upregulated differentiation genes may be indirectly regulated by CASZ1b through its upregulation in expression of MYOD, MYOG, or MEF2D. Our results indicate that CASZ1 both directly and indirectly regulates skeletal muscle differentiation genes to induce cell differentiation. Our findings are consistent with a model in which CASZ1 activates skeletal muscle genes and suppresses neural genes in ERMS cells, thus favoring skeletal muscle differentiation and restricting neuronal differentiation.

Consistent with being transcriptionally upregulated by CASZ1b, an increase of RNA Pol II signal within the gene body of CASZ1b bound upregulated genes (266 genes) was noted (Fig. 5c, left panel); although subtle, a decreased trend of RNA Pol II signal on the downregulated genes was observed (Fig. 5c, right panel). When focused on the CASZ1b peak center that was associated with upregulated genes, an increase of H3K27ac, RNA Pol II and ATAC-seq signal was observed after restoration of CASZ1b in SMS-CTR cells (Fig. 5d). Thus, highlighting that CASZ1b recruits co-activators to maintain open chromatin and upregulate gene transcription. For genes directly downregulated by CASZ1b (320 genes), there was a decrease of H3K27ac signal accompanied by a shape change within the CASZ1b peak center, but only a subtle decrease in the RNA Pol II and ATAC-seq signals (Fig. 5e). Interestingly, the ATAC-seq peak signal at the CASZ1b peak center is greater for upregulated genes than downregulated genes in CASZ1b over-expressed SMS-CTR cells (2.5 vs. 1.5 reads per million mapped counts) (Fig. 5d, e), indicating an increased chromatin accessibility for the upregulated genes. These findings indicate that restoration of CASZ1b levels in SMS-CTR cells leads to a more open chromatin and increased RNA Pol II loading on CASZ1-bound skeletal muscle gene loci to increase their expression (examples in Fig. 5f). Interestingly, unlike most upregulated skeletal muscle genes that showed an increase in H3K27ac and ATAC-seq signals, the decreases of H3K27ac and ATAC-seq signals were only observed at a few of the downregulated neural genes. For example, we observed decreases of H3K27ac and ATAC-seq signals at the *NGF* gene loci (Fig. 5g, left panel), but not at the *SOX*2 gene loci (Fig. 5g, right panel). Finally, we found that the loss of Casz1 led to a decrease of the signal intensities of both H3K27ac and H3K4me3 on skeletal muscle genes *Tnnt2*, *Mybph* and *Tnnc2* when normal muscle C2C12 cells were cultured in DM (Supplementary Fig. 5j).

**Restoration of CASZ1b in ERMS establishes new SEs**. Our study identified 499 SEs in control and 726 SEs in CASZ1b restored SMS-CTR cells (Fig. 6a and Supplementary Data 12), with over 600 of these SEs containing CASZ1b-binding sites. We observed increased H3K27ac signals ( > 1.4-fold) at the SE region of myogenic regulatory factors *MYOD*, *MYOG*, *MEF2D*, and skeletal muscle genes *TNNI1* and *TPM1* when CASZ1b was

restored (Fig. 6a). As expected, loss of Casz1 in normal muscle cells led to a decrease in the SE signals at *MyoD*, *Myog* and *Mef2d* (Fig. 6b). We found that there were 86 unique SEs in the control cells, 324 unique SEs in CASZ1b restored cells and 413 common SEs present in both conditions. Using a threshold of 1.4-fold, we found that 322 SEs increased H3K27ac signal and 36 SEs decreased H3K27ac signal upon expression of CASZ1 in SMS-CTR cells (Fig. 6c). IPA of the increased SE-associated genes showed enrichment in differentiation, formation of muscle, and proliferation of muscle cells (Fig. 6d). This is consistent with the previous findings that SEs determine cell fate, and the remodeling of SEs occurs throughout the course of cell differentiation[30,36]. For these SE driven genes, the increased signal of H3K27ac was accompanied by increased signal of RNA Pol II and ATAC (Supplementary Fig. 6a and Fig. 6e). Consistent with these findings, loss of Casz1 in differentiated C2C12 cells led to decreased H3K27ac binding on *MyoD*, *Myog* and *Mef2d* gene SEs (Supplementary Fig. 6b). Moreover, CASZ1b restoration in ERMS cells led to increases in MYOD, MYOG, MEF2D and endogenous CASZ1 expression (Fig. 6f).

**CASZ1 single-nucleotide variants (SNVs) discovered in human RMS**. We have shown that decreased expression of CASZ1 can occur in ERMS cells due to mutant RAS signaling (Fig. 2a–c). To determine whether there were other alterations in the CASZ1 gene that might have clinical implications, we examined sequencing data from 85 primary RMS tumors[9] and identified four tumors with nonsynonymous single-nucleotide variants (SNVs) in the CASZ1 gene that were either absent in the 1000 Genomes databases or had an extremely low-population frequency in other databases (Supplementary Fig. 7a). One single-nucleotide variant (c.73 C > T) identified from an ERMS patient tumor (with HRAS G12C mutation) resulted in the substitution of arginine to cysteine (R25C). The other SNVs identified from RMS result in the following amino acid (AA) changes: E323D, G676C, and M1129T. All of these SNVs are predicted to be deleterious (Supplementary Fig. 7a). R25C localizes to a nuclear localization signal (NLS) (AA 24–43) and G676C localizes to the zinc finger 4 C2H2 domain (ZF4) (AA 668–292) (Supplementary Fig. 7a, b). Both of these regions are highly evolutionarily conserved[26,28]. We generated CASZ1b mutant constructs of these SNVs and transfected them into HEK293T cells, and found that unlike wild-type (WT) CASZ1b or other CASZ1b SNVs, CASZ1bR25C localized to the cytoplasm (Fig. 7a), resulting in the loss of transcriptional activity when compared to wild-type CASZ1b in HEK293T cells (Fig. 7b). Thus, we focused on the R25C SNV. Using a Tet-inducible RDtetCASZ1bR25C cell line, we found that the R25C mutant had impaired nuclear localization, impaired transcriptional activity and decreased ability to suppress RD colony formation in soft agar (Fig. 7c–e). Importantly, an in vivo tumor xenograft model showed that while

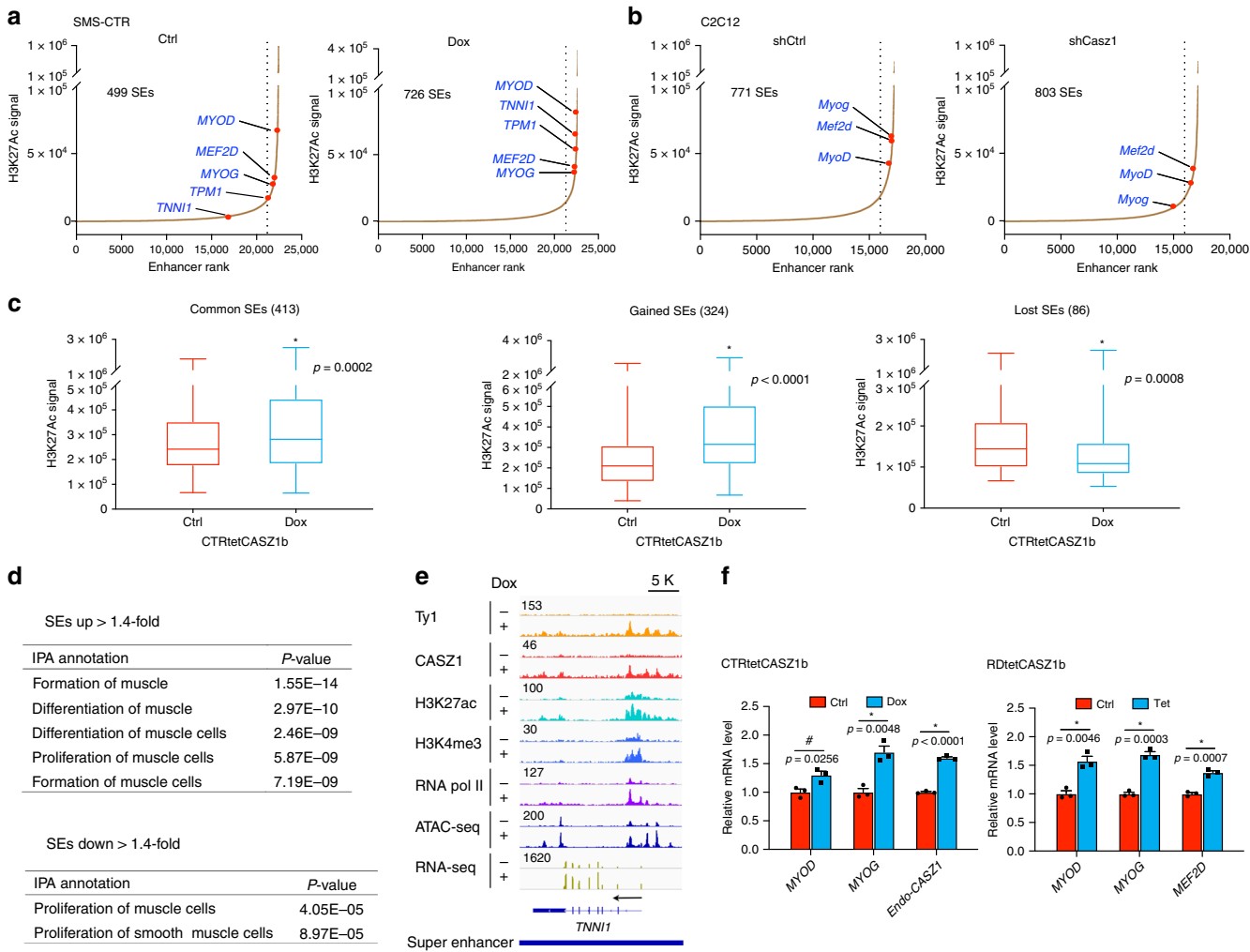

**Fig. 6 CASZ1b affects super-enhancers (SEs) establishment in ERMS. a** Restoration of CASZ1b in SMS-CTR cells increases SE numbers and leads to an increase of H3K27ac signal on MRFs and skeletal muscle genes. **b** Loss of Casz1 in C2C12 cells leads to a decrease of H3K27ac signal on MRFs but does not significantly affect SE numbers. **c** Restoration of CASZ1b in SMS-CTR cells leads to a significant increase of H3K27ac signal in gained SEs and common SE, and a decrease of H3K27ac signal in lost SEs. Data are presented as box and whisker plots with middle lines, indicating medians and whiskers representing the 25th and 75th percentiles. Two-sided Mann–Whitney test was used to calculate statistical difference. **d** In control or CASZ1b restored SMS-CTR cells, IPA analysis of genes driven by those SEs that with > 1.4-fold increase of H3K27ac signal are enriched in muscle formation, differentiation, and proliferation; SEs with > 1.4-fold decrease of H3K27ac signal are enriched in muscle proliferation. **e** Signal tracks show that restoration of CASZ1b in SMS-CTR increases SE signal (H3K27ac), as well as H3K4me3, RNA Pol II, ATAC-seq signal, and RNA-seq reads on the skeletal muscle gene *TNNI1*. **f** Restoration of CASZ1b in SMS-CTR cells increases MYOD, MYOG, and endogenous *CASZ1* mRNA level after 3 days Dox treatment compared to Ctrl (left panel). Restoration of CASZ1b in RD cells increases MYOD, MYOG, and MEF2D mRNA levels after 3 days Tet treatment compared to Ctrl (right panel). Data represent mean ± SEM, n = 3 biological replicates. Two-sided Student's t-test was used to calculate statistical difference. Source data are provided as a Source Data file.

overexpression of wild type CASZ1b significantly suppressed ERMS tumor growth and increased mouse survival, the CASZ1bR25C mutant had no effect on tumor growth (Fig. 7f and Supplementary Fig. 7d, e).

In summary, CASZ1 plays a critical role in inducing skeletal muscle and ERMS differentiation via forming a feed-forward loop with MYOD and MYOG; the loss of CASZ1 activity, due to RAS-MEK signaling or genetic alteration, impairs the formation of this core transcriptional regulatory circuitry and contributes to ERMS tumorigenesis (Fig. 7g).

## Discussion

In this study, we identify CASZ1 as a critical transcription factor involved in muscle differentiation and its dysregulation in ERMS contributes to their impaired myogenic program. We find that CASZ1 directly regulates MRFs, expression of skeletal muscle genes,

and myogenic differentiation in normal myoblasts and acts as a key component of the CRC that enables ERMS to differentiate when treated with MEK inhibitor. In ERMS, mutant RAS genes are capable of suppressing CASZ1 expression and rare genetic variants of CASZ1 are found in RMS, one of which is capable of inactivating CASZ1 transcriptional and tumor suppressor activity.

CASZ1 orchestrates cell specification and differentiation in many cell lineages, including neuroblasts, cardiomyocytes, and lymphoid cells[14,17,19,20,23,37]. CASZ1 cross-talks with other transcription factors to determine the ability of neuroblasts to generate progeny with distinct differentiation states in drosophila[18–22], and determine the retinal cell types in mice[16,38]. During skeletal myogenesis, MRFs are sequentially expressed and cross-talk with one another to encode the normal myogenesis process[11]. In this study, we show that CASZ1 cross-talks with MYOD and MYOG during myogenesis, whereas the loss of CASZ1 disrupts this myogenic program. Reminiscent of the role of Casz1 plays in drosophila neuroblasts

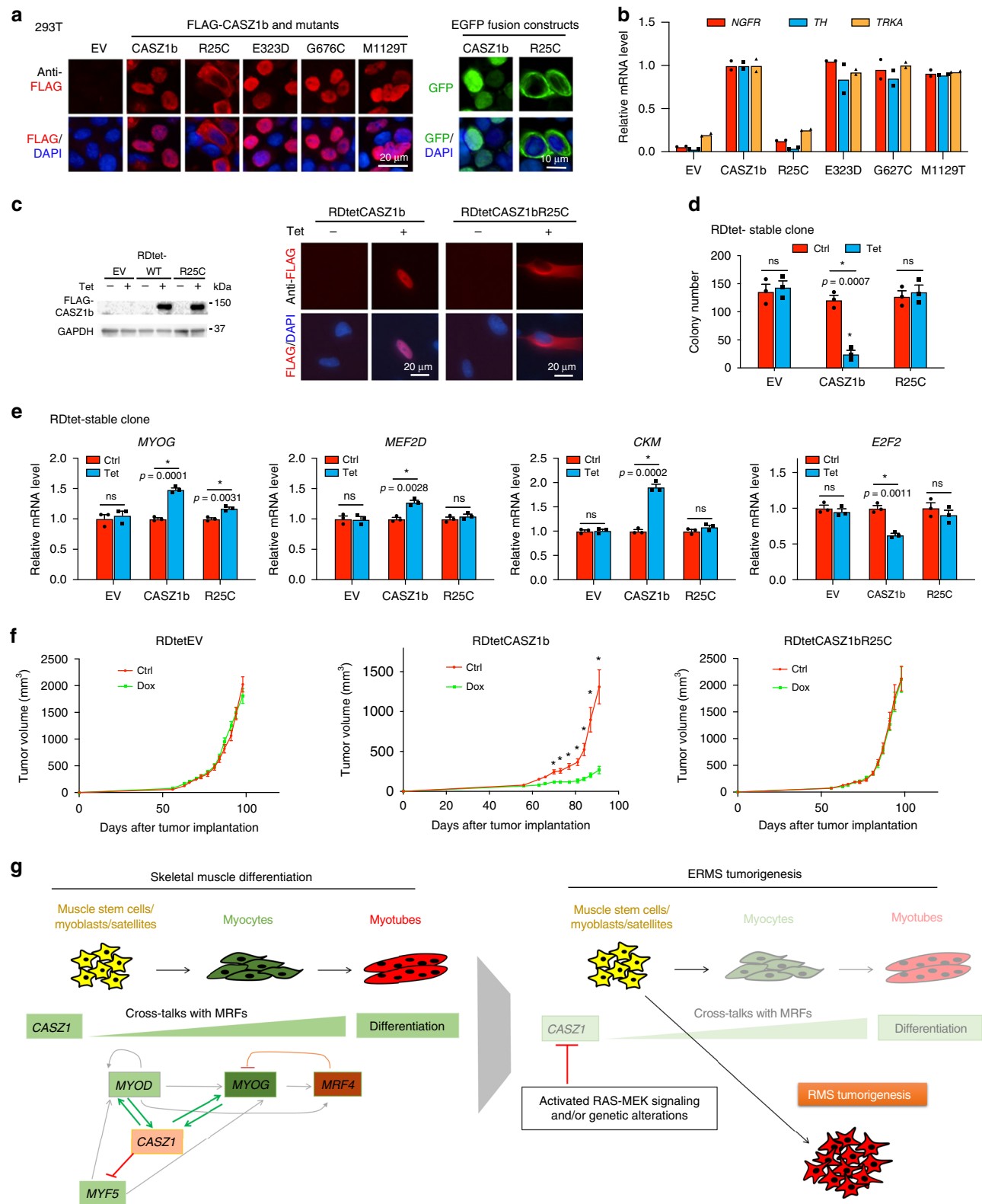

and mouse retina, our study indicates that CASZ1 cooperates with MRFs to regulate skeletal myogenesis. Taken together, our study suggests that CASZ1 regulates cell differentiation in different tissue types by cooperating with tissue-specific transcription factors.

For the first time, we define genome wide CASZ1-binding sites in a cellular differentiation system. We find that genes involved in a myogenic differentiation program are enriched in CASZ1 binding and the expression of these genes is upregulated by

CASZ1 (Fig. 5). There is not a significant positive correlation between increased global chromatin accessibility (ATAC-seq) and increased gene transcription (RNA-seq) upon induction of CASZ1 at the time point tested. CASZ1 binds to the promoter, enhancer and/or super-enhancer regions of MRFs and skeletal muscle genes, and the binding of CASZ1 on these regions is accompanied by increased chromatin accessibility and transcriptional activation as shown by increased ATAC-seq,

**Fig. 7 Identification of loss of function SNV in CASZ1 gene. a** Immuno-staining of HEK293T cells transiently transfected with FLAG-CASZ1b and FLAG tagged constructs of RMS SNVs using anti-FLAG antibody (red) and DAPI (blue) shows that wild-type CASZ1b and CASZ1b RMS SNVs E323D, G676C, M1129T localize to the nucleus of HEK293T cells, but CASZ1bR25C localizes to the cytosol (left panel). EGFP signal (green) and DAPI (blue) staining show that CASZ1b-EGFP fusion protein localizes to nuclei and while RMS SNV CASZ1bR25C-EGFP localizes to the cytoplasm (right panel). **b** CASZ1bR25C but none of the other RMS SNVs has decreased transcriptional activity compared to wild-type CASZ1b. **c** Western blots show the induction of wild-type CASZ1b and CASZ1bR25C in tetracycline inducible RDtetCASZ1b and RDtetCASZ1bR25C cells (left panel); immuno-staining of RDtetCASZ1b and RDtetCASZ1bR25C cells using anti-FLAG antibody (red) and DAPI (blue) indicates that CASZ1b localizes to the nucleus (DAPI, blue) but CASZ1bR25C localizes to the cytosol (right panel). **d** Restoration of wild-type CASZ1b but not mutant -R25C in RD cells (Tet) significantly suppresses soft-agar colony formation compared to controls. **e** Restoration of wild-type CASZ1b but not mutant R25C in RD cells significantly upregulates *MYOG, MEF2D, CKM* and represses *E2F2*. **f** Dox alone has no effect on tumor growth in empty vector (EV) transfected RD cells generated xenografts (left panel). Restoration of CASZ1b in RD cells (Dox, middle panel) significantly suppresses tumor growth in xenografts compared to controls. Restoration of CASZ1b R25C mutant in RD cells (Dox, right panel) does not affect tumor growth in xenografts compared to Ctrl. **g** Schematic diagram shows that CASZ1 regulates skeletal muscle differentiation via forming a feed-forward loop with MYOD and MYOG (left panel), while inactivation of CASZ1 due to activated RAS-MEK signaling or genetic alteration disrupts this feed-forward loop and results in ERMS tumorigenesis (right panel). Data represent mean, $n = 2$ biological replicates for **b**, data represent mean ± SEM, $n = 3$ biological replicates for **d**, $n = 10$ mice/group, *$p < 0.01$ for **f**, ns not significant. Two-sided Student's *t*-test was used to calculate statistical difference. Source data are provided as a Source Data file.

H3K27ac, RNA Pol II and RNA-seq signal within these regions (Fig. 5f, Fig. 6e, and Supplementary Fig. 6a). Thus, our results indicate that CASZ1 upregulates skeletal muscle genes through regional epigenetic modifications. CASZ1 also binds to neural genes in MEKi induced, differentiated SMS-CTR cells or upon restoration of CASZ1b expression in SMS-CTR cells, and this is associated with decreased expression of the neural genes in these muscle cells. CASZ1-mediated negative regulation of neural genes may reinforce the muscle cell identity during myogenesis. Our previous studies have shown that CASZ1 interacts with co-regulators PARP1 and NuRD complex[28,39]. Whether CASZ1 recruits these or other complexes to modify regional chromatin and regulate gene transcription in muscle cells warrants further investigation. Our data demonstrate that CASZ1 directly upregulates muscle genes in muscle cells and represses non-muscle genes by modifying the regional chromatin environment and SEs to favor skeletal muscle differentiation.

Our study shows that dysregulation of *CASZ1* is implicated in RMS tumorigenesis. We demonstrate that one mechanism by which CASZ1 is dysregulated in ERMS is through RAS-MEK signaling. Recent studies show that somatic mutations of *CASZ1* are significantly enriched in HPV-positive oral cancers[40,41], and *CASZ1* loss-of-function mutations are associated with heart disease[42,43], which indicates that genetic alterations in *CASZ1* are implicated in different types of diseases. With a limited number of patient RMS samples, we observed a loss-of-function genetic alteration in the *CASZ1* gene that contributes to the inactivation of CASZ1 transcriptional activities. In RMS, despite the binding of MYOD to the regulatory regions of its target genes, the majority of MYOD target genes remain under-expressed[1,12]. We find that CASZ1 forms a CRC with MYOD and MYOG in differentiated SMS-CTR cells, although we did not detect a physical interaction between CASZ1 and MYOD or MYOG. CASZ1 is a key component of this CRC, which is required for MRFs to be effective at upregulating their own expression as well as the expression of skeletal muscle genes necessary for muscle and ERMS differentiation. This is consistent with the critical role of a CRC at governing cell state and cell identity in other studies[33,44]. When CASZ1 was overexpressed in ERMS, we observed that CASZ1 directly binds to the regulatory regions of MYOD, MYOG, and MEF2D (Supplementary Fig. 6), and positively regulates their expression (Fig. 6f). Forced overexpression of MYOG or MEF2D suppresses RMS tumor growth and induces skeletal muscle differentiation[45,46]. These findings suggest a cooperative role of CASZ1 with MYOD, MYOG, and MEF2D to regulate myogenesis and facilitate ERMS cell differentiation.

Taken together, our results indicate that the regulation of muscle differentiation programs by CASZ1 is integral to proper myogenic differentiation. Either genetic alteration of CASZ1 or silencing of CASZ1 because of the activated RAS-MEK signaling disrupts the formation of a feed-forward loop of CASZ1, MYOD, and MYOG, which results in ERMS tumorigenesis and progression.

## Methods

**Cell culture.** Human embryonic kidney cells (HEK293T) and mouse C2C12 myoblasts were obtained from ATCC and were maintained in Dulbecco's modified Eagle's media (DMEM). ERMS SMS-CTR cells with HRAS mutation (HRAS_Q61K) were obtained from Dr. Peter Houghton, where the cell line was established. ERMS RD cells with NRAS mutation (NRAS_Q61H) and ARMS RH30 cells were obtained from Dr. Lee Helman's laboratory (Pediatric Oncology Branch, NCI), which were originally obtained from ATCC. RMS cells were maintained in RPMI1640 media. All the cell culture medium was supplemented with 10% fetal calf serum (FBS), as well as 100 μg/ml streptomycin, 100 U/ml penicillin, and ʟ-glutamine. Cells were grown at 37 ºC with 5% $CO_2$. The C2C12 cell differentiation medium (DM) contains DMEM and 2% horse serum. RMS differentiation medium (DM) contains RPMI1640, 10% FBS and 10 nM 12-O-tetradecanoylphorbol-13-acetate. Embryonal rhabdomyosarcoma RDtetEV, RDtetCASZ1b cells or RDtetCASZ1bR25C cells are RD cells that were first stably transfected with pcDNA6TR vector, then stably transfected with pT-REx-DEST30 empty vector or pT-REx-DEST30 containing N-terminal FLAG tagged CASZ1b or CASZ1bR25C mutant, and their expression is tetracycline (Tet) (1 μg/ml) inducible. Embryonal rhabdomyosarcoma SMS-CTRtetEV and SMS-CTRtetCASZ1b cells are CTR cells that infected with lentiviral particles generated using pLVX-TetOne-Puro empty vector or pLVX-TetOne-Puro vector containing C-terminal 3xTy1 tagged CASZ1b, and their expression is doxycycline (Dox, 0.5 μg/ml) inducible. Alveolar rhabdomyosarcoma RH30tetEV and RH30tetCASZ1b cells are RH30 cells that were first stably transfected with pcDNA6TR vector, then stably transfected with pT-REx-DEST30 empty vector or pT-REx-DEST30 containing N-terminal FLAG tagged CASZ1b, and their expression is induced by Tet (100 ng/ml) treatment. RDtetEV, RDtetCASZ1b, RDtetCASZ1bR25C, RH30tetEV, and RH30tetCASZ1b cell lines are single-clone selected and are cultured in the complete RPMI containing 40 μg/ml blasticidin and 500 μg/ml geneticin; SMS-CTRtetEV and SMS-CTRtetCASZ1b cell lines are single clone selected and are cultured in complete RPMI 1640 containing 1 μg/ml puromycin. Control or Casz1 stable knockdown C2C12 /SMS-CTR cell line was generated by infecting C2C12 cells or SMS-CTR with either control shRNA lentiviral particles (SigmaAldrich, MISSION pLKO.1-puro. catalog SHC002V) or CASZ1 shRNA lentiviral particles (SigmaAldrich, catalog TRCN0000131029) and selected using puromycin to generate pools. Wild-type HRAS, HRAS_Q61L, KRAS_G12D and NRAS_Q61K expression constructs were obtained from Addgene, and pBabe-containing retrovirus were produced to transduce C2C12 cells and generate RAS and RAS-mutant over-expressing stable clones. For C2C12 differentiation assay, differentiation index was calculated as the number of nuclei present in cells showing MHC staining, then divided by the total number of nuclei in the same field; fusion index was calculated as the number of nuclei present in cells showing MHC staining with greater than two nuclei divided by the total number of nuclei in the same field. For the differentiation assay, three independent fields (150–200 nuclei/field) were quantified. Cell confluency assay using Essen IncuCyte ZOOM or FLR was adopted to evaluate cell proliferation in realtime.

**Construction of missense and fusion CASZ1b constructs.** To make the CASZ1b-EGFP constructs, full length of CASZ1b was generated by PCR, and the primer used was synthesized with an attached 5′ *Eco*RI site and a 3′ *Bam*HI site and was cloned into the corresponding sites of pEGFP-N3. The QuikChange XL

Site-Directed Mutagenesis Kit (Agilent Technologies) was utilized to introduce various missense mutations into the wild-type FLAG-CASZ1b plasmid as per the manufacturer's protocol. To generate CASZ1b-EGFP mutant constructs, the FLAG-CASZ1b mutants were sub-cloned into CASZ1b-EGFP through SbfI and PshAI enzyme digestion.

**Transient transfection.** Transient transfection was performed as described previously[15]. siRNAs and allstars-negative control siRNA target different mouse genes are from Qiagen (Casz1 siRNA, Cat # 141261862; Myf5 siRNA, Cat # SI00178276 and SI02709749; MyoD siRNA, Cat # SI00196280 and SI04777745; Myog siRNA Cat # SI00229502 and SI02675918; Mek1 siRNA, Cat # SI01299613 and SI02668379, allstars-negative control siRNA, Cat # 1027281). siRNAs or plasmids are transiently transfected into C2C12 or RD cells using Nucleofector with solution V and program B-032 (Amaxa Biosystems), and plasmids are transfected into HEK293T cells using lipofectamine 2000.

**Real-time PCR.** Total mRNA was collected using the RNeasy Plus Mini Kit (Qiagen) as per the manufacturer's protocol. Quantitative measurements of total β-actin or Gapdh and other genes' levels were obtained using the BIO-RAD CFX Touch Real-time (RT) PCR detection system and performed in triplicate. Ct values were standardized to β-actin or Gapdh levels. Representative data from biological replicates were shown in this study. Primer sequences used for realtime PCR are shown below:

| Gene name | Forward primer | Reverse primer |
| --- | --- | --- |
| hACTA1 | CAGTTGTAGCTACCCGCCCA | CGTCGCCCACGTAGGAATCT |
| hCASZ1 | CAAAACAGACTCCATCACCACG | GTGCTGGCTGCCCGAGAAC |
| hCASZ1-Endogenous-specific | GATCCGGCAACTTTGGGCAG | GCAGGTGGGATGGAGGAGTC |
| hCKM | TTCCTGTTCGACAAGCCCGT | AAGGGGTGGCCAGCTTTCTT |
| hE2F2 | CGTCCCTGAGTTCCCAACC | GCGAAGTGTCATACCGAGTCTT |
| hMEF2D | CCAGCGAATCACCGACGAG | GCAGTCACATAGCACGCTC |
| hMYOD | CGCCATCCGCTATATCGAGG | CTGTAGTCCATCATGCCGTCG |
| hMYOG | AGTGCCATCCAGTACATCGAG | AGGTTGTGGGCATCTGTAGG |
| CCNB1 | AATAAGGCGAAGATCAACATGGC | TTTGTTACCAATGTCCCCAAGAG |
| CCNA2 | CGCTGGCGGTACTGAAGTC | GAGGAACGGTGACATGCTCAT |
| CDK1 | AAACTACAGGTCAAGTGGTAGCC | TCCTGCATAAGCACATCCTGA |
| PLK1 | AAAGAGATCCCGGAGGTCCTA | GGCTGCGGTGAATGGATATTTC |
| hNGFR | ACCTCATCCCTGTCTATTGCTCC | GCTGTTGGCTCCTTGCTTGTT |
| hTH | CCTACCAAGACCAGACGTACCAGTCA | TGCACCTAGCCAATGGCACTCA |
| hTNNC2 | TGGGGACATCAGCGTCAAG | CCAAGAACTCCTCGAAGTCGAT |
| hTNNI2 | GCCCTGCTGCCCAGATTCTA | CTCCTTCTCCAGCTCCGTGG |
| hTrkA | TTGGCATGAGCAGGGATATCT | ACGGTACAGGATGCTCTCGG |
| hβ-Actin | GCCAACCGCGAGAAGATGA | CATCACGATGCCAGTGGTA |
| mActa1 | GCCAGAGTCAGAGCAGCAGA | TTGCTCTGGGCCTCATCACC |
| mCasz1 | GCAGAAGAGCCCTCAAAAGATAA | GAAGCAGCGTAGTCCCTCAGA |
| mCkm | TTGACAAGCCCGTGTCACCT | CAGAAGCGGCGGAAAACCTC |
| mGapdh | AACTTTGGCATTGTGGAAGG | ATGCAGGGATGATGTTCTGG |
| mMyod | ACGCCATCCGCTACATCGAA | GTAGGCGGTGTCGTAGCCAT |
| mMyog | ATCCAGTACATTGAGCGCCTAC | GACGTAAGGGAGTGCAGATTGT |

**Antibodies.** The antibodies used for western blot, immunofluorescence staining, and ChIP-seq are obtained from different companies and amount needed is described as the following. Antibodies from EMD Millipore: MHC (Cat. # 05-716, 1:1000 for western blot; 1:50 for immunofluorescence), MYOD (Cat. # MAB3878, 1:1000 for western blot), MYOG (Cat. # MAB3876, 1:1000 for western blot), H3K27me3 (Cat. # 07-449, 4 μg/reaction for ChIP). Antibodies from Santa Cruz:

GAPDH (Cat. # sc-25778, 1:2000 for western blot), HRAS (Cat. # sc-520, 1:1000 for western blot), goat anti-mouse IgG HRP secondary antibody (Cat. # sc-2005, 1:2000), goat anti-rabbit IgG HRP secondary antibody (Cat. # sc-2004, 1:2000). Antibodies from Cell Signaling Technology: α-Tubulin (Cat. # 3873 s, 1:4000 for western blot), Total ERK ½ (Cat. # 9102, 1:1000 for western blot), p-ERK ½ (T202/Y204) (Cat. # 9106, 1:1000 for western blot), H3K4me3 (Cat. # 9751, 1:50 for ChIP), VINCULIN (Cat. # 4650 s, 1:10,000). Antibody from Abcam: H3K27ac (Cat. # ab4729, 4 μg/reaction for ChIP). Antibodies from Active Motif: Histone 3 (Cat. # 39163, 1:3000 for western blot), RNA polymerase II (Cat. # 39097, 4 μg/reaction for ChIP). Antibody from Diagenode: Ty1 (Cat. # C15200054, 1:2000 for western blot; 4 μg/reaction for ChIP). Antibody generated by collaborating with Rockland Immunochemicals Inc: CASZ1 (1:4000 for western blot, 4 μg/reaction for ChIP). Antibody from Molecular Probes: Goat anti-mouse IgG Alexa Fluor 594 (Cat. # A-11032, 1:250 for immunofluorescence staining).

**Protein isolation, co-immunoprecipitation, and western blot analysis.** For assessment of protein levels, cells were lysed using RIPA buffer, and 10 μg of total protein was separated and electroblotted as described previously[28]. Protein bands were detected using a goat anti-rabbit or mouse IgG-HRP conjugated secondary antibody (200 μg/ml; Santa Cruz Biotechnology) and visualized using enhanced chemiluminescence (Amersham Biosciences). Co-immunoprecipitation was performed as previously described[39]. In brief, rabbit anti-CASZ1 antibody was first incubated overnight with Dynabead M-280 Sheep Anti-Rabbit IgG magnetic beads (ThermoFisher Cat. # 11204D). Whole cell extracts from CTRtetCASZ1b cells treated with Dox for 24 h were incubated for 4 h with the CASZ1 antibody-bound magnetic beads under constant shaking at 4 ℃. The co-IP products were eluted by incubating with 1x SDS loading buffer heated to 100 ºC for 3 min. Protein bands were detected using a goat anti-rabbit IgG-HRP conjugated secondary antibody (200 μg/ml; Santa Cruz Biotechnology) and visualized using enhanced chemiluminescence (Amersham Biosciences).

**Immunofluorescence.** Cells were cultured in 8-well Lab-Tek Chamber Slides (Cat. no. 177402) for indicated time period. Cells were fixed, permeabilized, blocked, and stained as described previously[28]. For indirect immunofluorescent cell staining, an anti-FLAG M2 monoclonal antibody and an Alexa Fluor 594-conjugated goat anti-mouse antibody were used to detect FLAG-CASZ1b or mutant constructs. An anti-MHC antibody and an Alexa Fluor 594-conjugated goat anti-mouse antibody were used to detect MHC. Stained cells were imaged and analyzed using a Nikon Eclipse TE300 fluorescent microscope.

**RNA-seq.** Total RNA was isolated and subjected to RNA-seq analysis from C2C12 shCtrl and shCasz1 cells that were cultured in differentiation medium (DM) for 48 h; C2C12 cells that were transiently transfected with empty vector (EV), CASZ1a or CASZ1b, and cultured in regular growth medium (GM) for 48 h; RDtetCASZ1b cells that were treated with or without Tet for 48 h; and CTRtet-CASZ1b cells that were treated with or without Dox for 48 h. Total RNA extraction was carried out using an RNeasy Plus Mini Kit (Qiagen Inc.) according to the manufacturer's instructions. Strand-specific whole transcriptome sequencing libraries were prepared using TruSeq® Stranded Total RNA LT Library Prep Kit (Illumina, San Diego, CA, USA) by following the manufacturer's procedure. RNA-seq libraries were sequenced on Illumina HiSeq 2500 of paried-end with read length of 126 bp, or HiSeq 3000/4000 of paired-end with read lenth of 150 bp. The Fastq files with 126 bp or 150 bp paired-end reads were processed using Partek Flow. In brief, the raw reads are aligned using STAR and the aligned reads are quantified to annotation model through Partek E/M. The normalization method used here is counts per million (CPM) through Partek Flow. The normalized counts were then subjected to statistic analysis using GSA or ANOVA. To get T-score, the normalized counts acquired from Partek Flow are exported and further analyzed using Parteck Genomics Suite v7.17. To eliminate batch effect, some of the CPM got from Partek Flow were analyzed using DESeq2. Statistical results of differentially expressed genes from Partek Flow, or Parteck Genomics Suite v7.17 or DESeq2 were analyzed using QIAGEN's Ingenuity® Pathway Analysis (IPA®, QIAGEN) and gene-set enrichment analysis (GSEA) (http://www.broadinstitute.org/gsea/index.jsp). By default, the false-discovery rate (FDR) <0.25 is significant in GSEA. Heatmap of RNAseq results was generated using the ComplexHeatmap package in R 3.4.3. Previously published RNA-seq datasets generated from SMS-CTR cells that treated with or without MEKi trametinib (GSE85170, GSE85171)[10] were downloaded in this study. Linear correlation between expressions of the genes altered with CASZ1a and CASZ1b overexpressed C2C12 cells was done in R.

**ATAC-seq.** ATAC-seq was performed as previously described with subtle modifications[47]. In brief, SMS-CTRtetCASZ1b cells treated with or without Dox for 48 h. The trypsinized cells are resuspended in cold PBS. Pellet 100,000 viable cells in a 1.5 ml tube at 500 RCF at 4 ℃ for 5 min. Aspirate all supernatant, resuspend in 100 μl cold ATAC-Resuspension Buffer (RSB) containing 0.1% NP40, 0.1% Tween-20, and 0.01% Digitonin, incubate on ice for 3 min. Wash out lysis with 1.4 ml of cold ATAC-RSB containing 0.1% Tween-20, but no NP40 or digitonin (RSB-Wash Buffer). Pellet nuclei and resuspend cell pellet in 100 μl of transposition mixture (50 μl 2x TD buffer, 5 μl transposase (100 nM final), 33 μl PBS, 1 μl 1% digitonin,

1 μl 10% Tween-20, 5 μl H2O). Incubate reaction at 37 °C for 30 min in a ther-momixer at 1000 RPM. Cleanup reaction with a Zymo DNA Clean and Concentrator-5 Kit (cat# D4014). Elute DNA in 21 μl elution buffer, use 10–20 μl of product for library preparation. Nextera DNA Library Prep Kit (Illumina, catlog # FC-121-1030) was used for ATAC sequencing library preparation. After five cycles amplification using NEBNext 2x MasterMix, take 2.5 μl amplified ATAC library and perform a regular real-time PCR in duplicate using Fast SYBR (Thermo). The additional number of cycles needed for the remaining 45 μl PCR reaction is determined as follows: (1) Plot linear Rn vs. Cycle; (2) Determine the # of cycles that corresponds to 1/4 of maximum fluorescent intensity; (3) Add three additional cycles to the number determined in Step 2. In our case, we used 45 μl of the pre-amplified DNA, run three additional cycles without addition of any more reagents. The amplified library was right side selected using SPRIselect reagent (Beckman catlog # B23317) following their instruction.

ATAC libraries were sequenced on an Illumina NextSeq machine (2 × 75 cycles). The Fastq files with 75 bp paired-end reads were processed using Encode ATAC_DNase_pipelines (https://github.com/kundajelab/atac_dnase_pipelines) installed on the NIH biowulf cluster (https://hpc.nih.gov/apps/atac_dnase_pipelines.html). Previously published Dnase I-seq datasets generated from SMS-CTR cells that treated with or without MEKi trametinib (GSE85169, GSE85171)[10] were downloaded in this study.

**ChIP-seq**. ChIP was performed using the ChIP-IT High Sensitivity kit (Active Motif) as per the manufacturer's instruction. Briefly, formaldehyde (1%, 15 min) fixed cells were sheared to achieve chromatin fragmented to a range of 200–700 bp using an Active Motif EpiShear Probe Sonicator. C2C12 cells were sonicated at 30% amplitude, pulse for 20 s on and 30 s off for a total sonication "on" time of 14 min; SMS-CTR cells were sonicated at 20% amplitude, pulse for 20 s on and 30 s off for a total sonication "on" time of 13.5 min. Sheared chromatin samples were immunoprecipitated overnight at 4 °C with antibodies targeting H3K27ac, H3K4me4, H3K27me3, RNA Pol II, Ty1, CASZ1. Previous report showed that CASZ1 antibody could specifically recognize CASZ1a and CASZ1b[24]. To validate CASZ1 antibody that is feasible for ChIP-seq, we performed western blot analysis after immunoprecipitating the formaldehyde fixed cells using anti-CASZ1 antibody. We observed the pull-down of endogenous CASZ1a and CASZ1b in MEKi treated CTR cells, indicates this antibody is able to pull down crosslinked endogenous CASZ1 protein. Ty1 antibody is well established for ChIP-seq, here we performed ChIP-seq in 3-Ty1 tagged CASZ1b overexpressed CTR cells using either an anti-Ty1 antibody or an anti-CASZ1 antibody to further validate CASZ1 antibody for ChIP-seq. To normalize ChIP-seq signal, we employed Active Motif ChIP-seq spike in using Drosophila chromatin (Active Motif catalog # 53083) and an antibody against Drosophila specific histone variant H2Av (Active Motif, catalog #61686) according to the manufacturer's instructions. ChIP-seq DNA libraries were prepared by Frederick National Laboratory for Cancer Research sequencing facility. Libraries were multiplexed and sequenced using TruSeq ChIP Samples Prep Kit (75 cycles), cat. # IP-2-2-1012/1024 on an Illumina NextSeq machine. 25,000,000–30,000,000 unique reads were generated per sample. All the home generated ChIP-seq datasets can be found in the Gene Expression Omnibus (GEO) database.

**ChIP-seq data processing**. Previously published ChIP-seq datasets are downloaded for this study, which include ChIP-seq datasets of H3K27ac, MYOD, MYOG, RNA Pol II generated from CTR cells that treated with or without MEKi trametinib (GSE85169, GSE85171)[10]; ChIP-seq datasets of H3K27ac generated from HSMM and HSMM_tube (GSE29611)[48]; ChIP-seq datasets of H3K27ac, MyoD, Myog and Myf5 generated from C2C12_MB and C2C12_MT (GSE76010, GSE24852 and GSE44824)[49–51]. For the home generated ChIP-seq data, ChIP enriched DNA reads were mapped to reference human genome (version hg19) or mouse genome (mm10) using BWA[52]. Duplicate reads were infrequent but discarded. For IGV sample track visualization, coverage density maps (tdf files) were generated by extending reads to the average size (measured by Agilent Bioanalyzer minus 121 bp for sequencing adapters) and counting the number of reads mapped to each 25 bp window using igvtools (https://www.broadinstitute.org/igv/igvtools).

ChIP-seq read density values were normalized per million mapped reads. High-confidence ChIP-seq peaks were called by MACS2 (https://github.com/taoliu/MACS) with the narrow algorithm for TFs and broad peak calling for histone marks. The peaks that overlapped with the possible anomalous artifact regions (such as high-mappability regions or satellite repeats) blacklisted by the ENCODE consortium (https://sites.google.com/site/anshulkundaje/projects/blacklists) were removed using BEDTools. Peaks from ChIP-seq of CASZ1, Ty1, histone marks and RNA Pol II in cell lines were selected at a stringent p-value ($p < \text{1E-5} \sim p < \text{1E-7}$). Peaks within 1000 bp to the nearest TSS were set as promoter. The distribution of peaks (as intronic, intergenic, exonic, etc.) was annotated using HOMER. Enrichment of known and de novo motifs were found using HOMER script "findMotifsGenome.pl" (http://homer.salk.edu/homer/ngs/peakMotifs.html). Metagene plots of ChIP-seq and DNase data were performed using NGSplot[53].

The enhancers were identified using the ROSE2 (Rank Order of Super-Enhancers) software (https://github.com/BradnerLab/pipeline), using distal (> 2500 bp from TSS) H3K27ac peaks[36,54]. Enhancer constituents were stitched together if clustered within a distance of 12.5 kb. The enhancers were classified into typical and super-enhancers based on a cutoff at the inflection point in the rank ordered set (where tangent slope = 1) of the ChIP-seq signal (input normalized). $p < \text{E-7}$, broad-3 for CTRtetCASZ1b. $q < \text{E-5}$ for C2C12 shCtrl or shCasz1. For the core transcriptional regulatory circuitry analysis, the super-enhancer data and the binding motif of all transcription factors including newly defined CASZ1-binding motifs were uploaded and run through CRC mapper program (https://github.com/linlabcode/CRC)[33].

Among the 5408 CASZ1-binding sites, promoter regions were defined directly from the Homer annotated bed file within 1000 bp of the TSS. The remaining 5282 CASZ1-binding sites in the non-promoter regions were divided into enhancer-associated regions if they overlap with the enhancers (defined by H3K27ac ChIP-seq and enhancer analysis from ROSE) and others if they do not overlap with enhancers. The analyses were done in R using package ChIPpeakAnno.

To determine the alterations on super enhancers when CASZ1 is restored in SMS-CTR cells, signal intensity of H3K27ac ChIPseq +/−20K of super-enhancer peak centers was extracted and compared between control and Dox treated cells. Briefly, bed files of all super-enhancers were generated by Rose using K-mean ranked ChIP results of H3K27ac. ComputeMatrix function of the deepTools was used to generate a matrix of signal intensity of H3K27ac up- and down-stream of the super-enhancer centers, as intensity scores in 1000 bp bins. The matrix of signal intensity was further used in R (version 3.5.2) to calculate the accumulated signal around each super-enhancer center to be used as signal intensity for each super-enhancer.

Pearson correlation coefficient was calculated for the pairwise comparison between Ty1 and CASZ1 ChIP samples using deepTools. Firstly, multiBigwigSummary was used to binarize the signal intensity of bigwig files of the two samples, based on the interested regions specified in the provided bed file (CASZ1 peak file in Dox treated CTR cells here). A matrix of the average score of both samples in each bin of our interested regions was the output of multiBigwigSummary, which was used next for the pairwise comparison. Finally, plotCorrelation was used to perform the Pearson correlation analysis and the pairwise scatterplot was also generated.

Heatmaps of signal intensity of ChIP samples were generated using deepTools. Briefly, computeMatrix was used to calculate signal intensity scores per ChIP sample in a given genome region that was specified by a bed file. The output of computeMatrix was a matrix file of scores of two ChIP samples, which was then used to generate the heatmap using plotHeatmap.

**Xenograft tumor studies**. Four- to 6-weeks-old female Fox Chase SCID Beige (CB17.B6-Prkdc$^{scid}$Lyst$^{bg}$/Crl) mice were ordered from charles river (Wilmington, MA) were housed on a 12-12-h light–dark cycle at 22 °C with free access to lab chow and water. $N = 20$ mice for each cell line were randomly divided in two groups, one group of mice received normal chow diet (Teklad/Envigo), while the other group has received Dox-containing chow diet (Bio-Serv) for over 5 days and continue to be fed with Dox-containing chow after cell injection. Subconfluent RDtetEV, RDtetCASZ1b, and RDtetCASZ1bR25C were harvested by trypsinization and resuspended in Hank's balanced salt solution (HBSS/Invitrogen), and Matrigel (Trevigen, Gaithersburg, MD, USA) at a (1:1) final concentration of $2 \times 10^7$ cells/ml. All mice were injected with $2 \times 10^6$ in 100 μl cell suspension into the left hind limb at the gastrocnemius muscle groups of the SCID Beige mice. In all experiments, the leg dimensions were measured twice a week with digital calipers to obtain two diameters of the tumor diameter, from which the tumor volume was determined using the equation $(D \times d2)/6 \times 3.14$ (where $D$ = the largest diameter) and ($d$ = the smallest diameter). All mice were euthanized when any of the tumor diameters were approaching 20 mm in any dimension. The log-rank (Mantel-Cox) test was used to compare event-free survival distributions between treatment groups. All Xenograft studies were approved by the National Cancer Institute's Animal Care and Use Committee (ACUC), and all animal care was in accordance with institutional guidelines.

**Statistics and reproducibility**. The statistical analyses used throughout this paper are specified in the appropriate results paragraphs and Methods sections. Additional statistical analyses were performed using Microsoft Excel, standard two-tailed Student's t-test, one-way ANOVA, and the software GraphPad Prism 8.1.0. Representative experiment such as micrographs has been repeated at least two to three times.

**Reporting summary**. Further information on research design is available in the Nature Research Reporting Summary linked to this article.

## Data availability

All the home generated RNA-seq, ChIP-seq, and ATAC-seq datasets can be found in the Gene Expression Omnibus (GEO) database. GEO accession number for data generated in this study is GSE126147. GEO accession numbers for publicly available ChIP-seq and RNA-seq data are GSE85169, GSE85171, GSE29611, GSE76010, GSE24852, and GSE44824. To evaluate CASZ1 mRNA levels in RMS patients, we queried microarray data deposited in R2 database (https://hgserver1.amc.nl/cgi-bin/r2/main.cgi) and RNA-seq data from the Integrated Rhabdomyosarcoma Databases (iRDb, the Childhood Solid Tumor Network at St. Jude, https://www.stjude.org/research/resources-data/childhood-solid-tumor-network/available-resources.html#irdb). The source data underlying Fig. 1a,

c–f, h, j, 2a–c, 3f, g, 4a–c, 5a, b, 6c, f, 7b–f and Supplementary Figs. 1c, d, 2d–g, i–l, 4a–c, e–g, 7c–e are provided as a Source Data file.

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

## Acknowledgements

This work was funded by the Center for Cancer Research, Intramural Research Program at the National Cancer Institute. We thank Dr. Deblina Benerjee for the advice on bioinformatic analyses. We thank Drs. Ming Sun and Man Xu for

insightful discussions. We thank Dr. Lee J. Helman and Mr. Choh Yeung from National Cancer Institute for providing the RD and RH30 cell lines that were stably transfected with pcDNA6/TR vector. We thank the Childhood Solid Tumor Network at St. Jude for providing normal and RMS RNAseq data. We thank Bao Tran, Jyoti Shetty, and Yongmei Zhao from NCI Sequencing Facility for ChIP-DNA and RNA sequencing.

## Author contributions

Z.L. and C.J.T designed the study, wrote the manuscript, and coordinated the entire study. Z.L. performed the in vitro and in vivo experiments and analyzed the data. X.Z. performed the ChIP experiments. X.Z., Z.L., and H.L. performed bioinformatic analyses. N.L. helped with generating CASZ1 overexpression stable clones, their analysis, and western blots. O.Y., M.E.Y., and E.R.H helped with MEK inhibitor treatment experiments. S.C. and M.X. helped on cell culture, realtime PCR, and western blots. A.M. helped on animal experiments. J.F.S., J.S.W., and J.K. identified the SNVs in the CASZ1 gene from RMS patients. X.Z., H.L., and A.M. contributed to the writing of the methods part of the manuscript. J.F.S., M.E.Y., X.Z., J.K., and J.S.W. edited the manuscript. All the authors have reviewed the manuscript.

## Competing interests

The authors declare no competing interests.
