## [Peer Review File · Nature Communications]

Reviewers' comments:

Reviewer #1 (Remarks to the Author):

In embryonal rhabdomyosarcoma (ERMS), cells are unable to undergo myogenic differentiation despite MYOD expression, mechanism of which remains unclear. In this study, Liu et al. tried to uncover the underlying mechanisms mainly using genome-wide approaches and found the zinc finger transcription factor CASZ1 as a critical factor regulating cell-fate decision in ERMS. CASZ1 expression was shown to be negatively regulated through the RAS-MEK pathway, which is aberrantly activated in ERMS. NGS-based experiments indicated that CASZ1 globally upregulates skeletal muscle-related genes in part in coordination with myogenic regulatory factors like MYOD while epigenetically suppresses non-myogenic genes including neural genes. In addition, the authors identified a patient-derived mutation of CASZ1 that attenuates its function. Many of these results, however, do not provide direct evidence for the regulatory mechanisms, although they are of potential interest for readers of the journal. The authors need to more directly address several points, e.g., as listed below, to strengthen the findings.

Major points

Page 4, Fig. 1: The authors stated that "CASZ1 is directly regulated by myogenic regulatory factors (MRFs) in myoblasts", but evidence for "direct regulation" is missing. In Fig. 1b, the authors should perform peak-calling for each factor to demonstrate their bindings on the Casz1 gene. An alternative would be to perform ChIP-qPCR for both MyoD and MyoG on the locus.

Page 7, Fig. 2: ChIP-Seq identified 5408 (Page 7) or 5406 (legend for Fig. 2h) CASZ binding sites, 2 % and 90% of which were at promoters and within intronic or intergenic regions, respectively. However, it is unclear how many binding sites are within gene regulatory regions like enhancers. This point should be indicated in Fig. 2h.

Page 8, Fig 3: What is the definition of "a component of the core transcriptional regulatory circuitry (CRC)"? Does it need to directly interact with other CRC components for cooperative actions? Quantitative criteria, as well as if CASZ1 satisfies those as a CRC component, should be clearly mentioned.

Pages 13-14, Fig. 7: Due to the limited number of data analyzed, the results provided here do not sufficiently demonstrate that the identified mutation R25C is clinically relevant. Accordingly, the sentence in the Abstract, "Next generation sequencing of primary RMS tumors identified a single nucleotide variant in the CASZ1 coding region that contributes to ERMS tumorigenesis", appears overstated.

Minor points

Page 5, Fig. 1: Which isoform, Casz1a or Casz1b, is dominantly expressed in skeletal muscle? Western blot results in Fig. 1a and Fig. 1d-f appear opposite. In addition, what is the difference between two isoforms (e.g., presence or absence of domains)? More details should be provided e.g. by citing references.

Page 5, Fig. 1i: The authors claimed "no detectable isoform specificity between CASZ1a and CASZ1b," but Fig. 1i appears to indicate that CASZ1a affects the differentiation more. These tendencies are also found in Sup. Fig. 1c-d, although differences between isoforms may not be statistically tested. Longer differentiation induction (more than 48 hr) would provide clearer results.

Fig. 1j: The legend should be "Knockdown Casz1 in C2C12 cells cultured in DM for 24 hr leads to INCREASES in Myf5 mRNA levels and DECREASES in MyoD and Myog mRNA levels".

Pages 6-7, Fig. 2c: The authors used patient-derived RAS mutation constructs but not the wild-

type one. The authors may need to include the wild-type for comparison and/or provide evidence that the mutants used here are active.

Pages 7-9, Figs. 2d-h and 3a-e: It is not clear how the genome-wide binding of CASZ1 is determined. Since CASZ1 is shown to play a role in differentiation systems other than myogenic one (as described in the Introduction and Discussion sections), there should be tissue/cell type-specific mechanisms, which might involve myogenic transcription factors such as MYOD. The authors should more discuss a potential mechanism on this point: e.g., if CASZ1 interacts with MYOD or other transcription factors, if CASZ1 binding is upstream or downstream of MYOD and/or MYOG binding (Fig. 3b-d for instance indicates MEKi treatment appears to increase MYOG, but not MYOD, binding), how “a cooperative role of CASZ1 with MYOD, MYOG and MEF2D” (Page 17) is regulated.

Pages 10-13, Fig. 5: Related to the point above, the molecular roles of CASZ1 on genome are not well documented. CASZ1 not only binds to myogenic genes for the formation of active chromatin environments, but also occupies neural genes to somehow negatively regulate their expression. These opposite actions of CASZ1 are intriguing but may need to be a bit more interpreted or molecularly dissected, although this point is discussed in the Discussion section (Page 16). For instance, the authors could more analyze chromatin accessibility (ATAC-seq) data in Fig. 5 by detecting differentially accessible regions to identify specific genomic loci/genes that are positively or negatively regulated by CASZ1.

There are some typos throughout the manuscript (e.g., middle of Page 13, “differentiation and proliferation of muscle cells of muscle cells”). Please check the text carefully again.

Reviewer #2 (Remarks to the Author):

In this study, Liu and colleagues explore the functional role of the Casz1 transcription factor in rhabdomyosarcoma (RMS). Since RMS is related to a skeletal myogenic lineage, the authors examine the expression and regulation of Casz1 in C2 myoblasts. They show that Casz1 is highly induced during differentiation and regulated by myogenic transcription factors MyoD and myogenin, the same factors whose functions are impaired in RMS. Gain or loss of Casz1 expression in C2 myoblasts supported that Casz1 functions as a positive regulator of skeletal muscle differentiation, which the authors further showed was mediated through the control of MyoD and myogenin expression and their downstream gene targets. Thus, Casz1 and the myogenic transcription factors operate in a regulatory loop to control myogenesis.

Next, the authors investigated the relevance of Casz1 in RMS. They particularly focus on the ERMS subtype and find that Casz1 was downregulated in RD cells. Treatment of RD cells with a MEK inhibitor promoted differentiation, concomitant with increases in Casz1, which was reversed with expression of activating mutants of KRas.

By ChIP-Seq analysis of Casz1 binding to chromatin, data showed that enrichment of Casz1 predominated in differentiated ERM cells, at gene sites containing E box consensus sequences and overlapping with enrichment peaks identified with active transcription marks. The same E box sites were predominantly occupied by myogenic transcription factors, MyoD and myogenin. These data suggested that inhibition of Ras-MEK signaling induced differentiation in ERMS, was mediated by cooperative function between Casz1 and myogenic transcription factors, similar to how C2C12 differentiation was controlled.

To assess the human tumor relevance, the authors screened RMS tumors and found a small subset containing Casz1 mutations. They analyzed one of these mutants containing an R25C variant, which was shown to cause the cellular mis-localization of Casz1 and reduce transactivation of

myogenic genes. In vivo, the injection of an ERMS cell line expressing the Casz1 R25C variant in mice had no effect on tumor growth as opposed to wt Casz1 which suppressed tumor growth.

The authors conclude that similar to MyoD, Casz1 functions in ERMS as a tumor suppressor by acting cooperatively with myogenic transcription factors to promote differentiation of muscle progenitor cells.

Critique

Overall, this is carefully performed study that provides insight into the function of Casz1 and its relation to RMS and skeletal muscle differentiation. There are no major limitations in the study, but the points below refer to areas in the manuscript that were not sufficiently addressed.

1. It was unclear whether Ras-MEK signaling is sufficient to regulate Casz1 expression. The authors should perform some gain and loss of function experiments in C2C12 myoblasts to address whether Ras-MEK signaling can regulate Casz1.

2. It was also unclear why the authors solely focused on ERMS. They should check whether Casz1 expression levels are altered in ARMS. Given that MyoD function is impaired in both ERMS and ARMS subtypes, and that the authors showed that MyoD regulates Casz1, one would suspect that Casz1 might also be relevant in the more aggressive form of RMS.

3. The relevance of the findings would be significantly enhanced if evidence of Casz1 regulation was confirmed in ERMS patient samples (by qRT-PCR or IHC).

Minor comments

a. The authors mentioned using an RMS integrated database, but the identity and details of this resource were not described in the methods section.

b. Line 209: the results in this sentence are confusing since Casz1 expression would be expected to be lower in ERMS.

Reviewer #3 (Remarks to the Author):

In this manuscript, Liu et al characterized the role of CASZ1, a zinc finger transcription factor, in skeletal muscle and embryonal rhabdomyosarcoma (ERMS) differentiation by genetic (knockdown and overexpression) and integrative genomic studies. They first showed that CASZ1 is required for the upregulation of MYOD signature genes and skeletal muscle differentiation in normal C2C12 myoblast cells and ERMS cell lines. They noted that the expression of oncogenic RAS proteins impaired the expression of CASZ1 in ERMS cells. Then by integrative analysis of changes in gene expression (by RNA-seq), chromatin accessibility (by ATAC-seq), chromatin occupancy of key TFs and histone marks (by ChIP-seq), they provided evidence that CASZ1 directly upregulates skeletal muscle related genes and represses non-muscle genes such as neural lineage related genes. CASZ1 also forms a core regulatory circuitry with MYOD and MYOG in an autoregulatory loop to control enhancer and super-enhancer activity. Finally, they identified several genetic variants affecting CASZ1 coding sequences in ERMS tumor samples, and provided evidence that one of them (R25C) results in impaired CASZ1 cellular localization as the possible mechanism in RMS tumorigenesis.

Overall, this study presents a set of interesting findings and a nice synthesis of the various epigenomic profiling results in myoblasts and ERMS cells with CASZ1 manipulations. The initial characterization of CASZ1 in normal myoblast differentiation was rigorous and the results were carefully discussed. The analysis of RNA-seq, ATAC-seq and ChIP-seq datasets, and the integrative

analysis of multiple datasets were thorough and well executed. The detailed analysis of enhancer and super-enhancer signatures upon CASZ1 depletion or restore expression in ERMS cells provides new insights into the genomic mechanisms by which CASZ1 regulates the myogenic differentiation programs in normal myoblasts and ERMS cells. The identification and characterization of genetic variants affecting CASZ1 function in ERMS patients is an important finding that provides a mechanistic explanation for at least one genetic variant R25C that impairs the tumor suppressive function of CASZ1 in ERMS. In this regard, this work represents an important advance in our understanding of this important transcription factor in the context of myogenesis and ERMS tumorigenesis. There are several questions remain, that if addressed to further improve the already developed manuscript, this work may have an excellent impact.

Major points:

1. The authors provide convincing evidence that activated RAS-MEK signaling by overexpression of oncogenic RAS leads to the suppression of CASZ1 expression in ERMS cell lines. Since oncogenic RAS-MEK also inhibits skeletal muscle differentiation, whereas inhibition of RAS-MEK (i.e. by MEKi or MEK1 knockdown) induces skeletal muscle differentiation, it is important to determine whether the suppression of CASZ1 expression by RAS-MEK is due to direct or indirect effect. This is relevant because the expression of CASZ1 is significantly and progressively upregulated during skeletal muscle differentiation, it is likely that the observed effect on CASZ1 expression was due to impaired differentiation rather than direct regulation of CASZ1. Also, is the suppressive effect on CASZ1 by RAS-MEK specific to myoblasts or ERMS cells or is it also seen in other cellular contexts? It would be interesting to determine whether mutant RAS also suppresses CASZ1 expression in cell/tumor types where CASZ1 is also known to be important and RAS mutations are commonly found. This line of investigation may help address the above question whether the RAS-MEK-mediated suppression of CASZ1 is due to direct or indirect mechanisms.

2. Along the same line, when discussing gene expression changes upon CASZ1 modulations (i.e. knockdown, overexpression or restored expression), it would be helpful to further clarify whether the observed changes are likely due to direct effects by CASZ1-mediated transcriptional control or indirect effects due to impaired cell differentiation. Since many of the affected genes or gene signatures are related to cell differentiation, and only a small subset of differentially expressed genes had CASZ1 chromatin binding by ChIP-seq (Fig. 5), it seems that most of the observed gene expression changes are due to indirect effect on myoblast differentiation.

3. Most of the integrative analysis of RNA-seq, ChIP-seq and ATAC-seq were performed in the context of ERMS cells, although the authored did analyze gene expression changes (by RNA-seq) and epigenetic changes (by ChIP-seq) in C2C12 myoblast cells (Fig. 1 and Supplementary Fig. 5j). It would be helpful to spend some efforts to have more in-depth analysis of CASZ1-mediated transcriptional programs using these datasets, and compare the results with that in ERMS cells. It would be interesting to know whether there are shared or specific programs that are regulated by CASZ1 in normal myoblast differentiation vs ERMS rhabdomyosarcoma differentiation, which will help elucidate the function of CASZ1 in normal vs tumor cells.

Minor points:

1. In Supplementary Fig. 5f, the authors concluded that restoration of CASZ1b in SMS-CTR cells did not significantly affect global ChIP-seq and ATAC-seq signals, whereas in Fig. 5d,e the correlative changes in H3K27ac, RNA Pol II and ATAC-seq were observed at CASZ1b bound and differentially expressed genes. This is somewhat confusion and should be clearly stated that different genes or genomic regions were used for these analyses.

2. It would be helpful to also comment on possible functions of other genetic variants identified in this study. Are they predicted as deleterious mutations? Do they impact specific protein domains and/or conserved residues?

III. Response to the reviewers' comments are under each of the comments and highlighted in blue:

Reviewer #1 (Remarks to the Author):

In embryonal rhabdomyosarcoma (ERMS), cells are unable to undergo myogenic differentiation despite MYOD expression, mechanism of which remains unclear. In this study, Liu et al. tried to uncover the underlying mechanisms mainly using genome-wide approaches and found the zinc finger transcription factor CASZ1 as a critical factor regulating cell-fate decision in ERMS. CASZ1 expression was shown to be negatively regulated through the RAS-MEK pathway, which is aberrantly activated in ERMS. NGS-based experiments indicated that CASZ1 globally upregulates skeletal muscle-related genes in part in coordination with myogenic regulatory factors like MYOD while epigenetically suppresses non-myogenic genes including neural genes. In addition, the authors identified a patient-derived mutation of CASZ1 that attenuates its function. Many of these results, however, do not provide direct evidence for the regulatory mechanisms, although they are of potential interest for readers of the journal. The authors need to more directly address several points, e.g., as listed below, to strengthen the findings.

Major points

Page 4, Fig. 1: The authors stated that “CASZ1 is directly regulated by myogenic regulatory factors (MRFs) in myoblasts”, but evidence for “direct regulation” is missing. In Fig. 1b, the authors should perform peak-calling for each factor to demonstrate their bindings on the Casz1 gene. An alternative would be to perform ChIP-qPCR for both MyoD and MyoG on the locus.

We appreciate the reviewer’s insights. In the original Fig. 1b signal tracks we showed that MyoD and Myog bind to Casz1 gene locus in C2C12 cells. However, in the original data set (GSE49313) there was no input, which is required for peak calling. For this reason, we analyzed a different MyoD and Myog ChIP-seq data that includes the input control (GSE44824). This time we performed a peak calling ($p < 10^{-7}$) and identified around 20 peaks on CASZ1 gene locus for both MyoD and Myog. The called peaks are marked under the signal tracks in the revised Fig. 1b and detailed information can be found in revised Suppl. Table 1.

Page 7, Fig. 2: ChIP-Seq identified 5408 (Page 7) or 5406 (legend for Fig. 2h) CASZ binding sites, 2 % and 90% of which were at promoters and within intronic or intergenic regions, respectively. However, it is unclear how many binding sites are within gene regulatory regions like enhancers. This point should be indicated in Fig. 2h.

We appreciate the reviewer's comments and pointing out the discrepancy between the number of ChIP-Seq peaks. We have changed the typo 5406 to 5408 in legend of Fig. 2h. To identify how many CASZ1 binding sites are at enhancers, we overlaid the H3K27ac binding sites that were identified as enhancers based on ROSE2 software (see methods section in the manuscript) with CASZ1 binding sites in trametinib treated SMS-CTR cells. We have included a color key to identify enhancers, promoter and unassigned regions in the revised Fig. 2h. We found that 2771 CASZ1 binding sites (51.2%) are within enhancers. We described this observation on page 9.

Page 8, Fig 3: What is the definition of “a component of the core transcriptional regulatory circuitry (CRC)”? Does it need to directly interact with other CRC components for cooperative actions? Quantitative criteria, as well as if CASZ1 satisfies those as a CRC component, should be clearly mentioned.

We have clarified the definition of CRC on page 10 by inserting the following: “*CRC is comprised of a group of transcriptional factors (TFs), marked by the presence of super-enhancers (SEs). These CRC TFs not only bind to their own gene loci and regulate their own gene expression, and also mutually regulate each other member in the CRC, thus forming an interconnected auto-regulatory feed-forward loop³⁷. CASZ1 is a component of the CRC together with MYOD and MYOG in differentiated SMS-CTR cells, based on (1) CASZ1 expression is regulated by itself (Supplementary Fig. 1d) and is driven by a SE (Supplementary Fig. 2a,h); (2) CASZ1, MYOD and MYOG bind consensus DNA sequences adjacent to each other within their own SE regions (Fig. 3b-d); (3) CASZ1, MYOD and MYOG positively regulate each other's expression in muscle cells (Fig. 1c,j, Supplementary Fig. 1d); and (4) among ~700 SEs in the differentiated SMS-CTR cells, over 80% contain CASZ1, MYOD and MYOG binding motifs based on motif scan using the CRC analysis tool³⁷”.*

Published studies of CRC have never shown that a direct interaction between each component is required for their cooperative actions. Moreover, ChIP-seq doesn't resolve time – i.e. a region can be bound by multiple factors not all at the same time or in the same cell but that would still show up as an overlapping peak. However, it is still an interesting question to see whether there is an interaction between CASZ1 and MRFs. To answer this question, we performed a co-immunoprecipitation in CASZ1b-restored SMS-CTR cells using an anti-CASZ1 antibody, and we did not observe an interaction between CASZ1 and MYOD or MYOG (revised Suppl. Fig. 4g). We described this observation on page 12.

Pages 13-14, Fig. 7: Due to the limited number of data analyzed, the results provided here do not sufficiently demonstrate that the identified mutation R25C is clinically relevant. Accordingly, the sentence in the Abstract, “Next generation sequencing of primary RMS tumors identified a single nucleotide variant in the CASZ1 coding region that contributes to ERMS tumorigenesis”, appears overstated.

We appreciate the reviewer's comments and have revised this sentence. In the abstract we changed the last sentence to “Next generation sequencing of primary RMS tumors identified a single nucleotide variant in the CASZ1 coding region that **potentially** contributes to ERMS tumorigenesis” by adding ‘potentially’.

Minor points

Page 5, Fig. 1: Which isoform, Casz1a or Casz1b, is dominantly expressed in skeletal muscle? Western blot results in Fig. 1a and Fig. 1d-f appear opposite. In addition, what is the difference between two isoforms (e.g., presence or absence of domains)? More details should be provided e.g. by citing references.

Usually the Casz1a and Casz1b isoforms are co-expressed and proteins are found to be expressed at similar levels (see the updated western blot results in Fig. 1d,e). We have included more details on these two isoforms in the introduction section (page 3) by inserting the following: “CASZ1 gene encodes two major isoforms. Human CASZ1a, has 1759 amino acids (AA) with 11 TFIIIA class C2H2 zinc fingers while CASZ1b is the more evolutionarily conserved isoform and is comprised of the first 1166 AA of CASZ1a but lacks the last 6 zinc fingers¹⁵. Both CASZ1a and CASZ1b function similarly to suppress neuroblastoma growth and regulate expression of neuronal genes²⁴⁻²⁶. However, Casz1 isoforms have been shown to play distinct roles in murine retina progenitor cells¹⁶, with overexpression of Casz1a favoring the development of bipolar cells while overexpression of Casz1b favors the production of rod photoreceptor cells¹⁶”.

Page 5, Fig. 1i: The authors claimed “no detectable isoform specificity between CASZ1a and CASZ1b,” but Fig. 1i appears to indicate that CASZ1a affects the differentiation more. These tendencies are also found in Sup. Fig. 1c-d, although differences between isoforms may not be statistically tested. Longer differentiation induction (more than 48 hr) would provide clearer results.

As suggested by the reviewer, we have tried to do longer differentiation induction (72 hr) and the results were not clearer. We believe this is due to the nature of the experiments with transient transfections. We have though, extended our analysis of the q-PCR depicted in Suppl. Fig 1d to include the entire transcriptomes (RNA-seq) regulated by the different isoforms when CASZ1a or CASZ1b are overexpressed in C2C12. We find that the genes regulated by each isoform are significantly-positively correlated, which suggests that these two isoforms have a similar activity at regulating gene transcription in C2C12 cells. We have included this data in revised Suppl. Fig. 1f and Suppl. Table 3. We believe at this level of analysis the effects of the different isoforms are more similar than they are different.

Fig. 1j: The legend should be “Knockdown Casz1 in C2C12 cells cultured in DM for 24 hr leads to INCREASES in Myf5 mRNA levels and DECREASES in MyoD and Myog mRNA levels”.

Thank you for pointing out the error. We have corrected this sentence.

Pages 6-7, Fig. 2c: The authors used patient-derived RAS mutation constructs but not the wild-type one. The authors may need to include the wild-type for comparison and/or provide evidence that the mutants used here are active.

To address this question, we transduced C2C12 cells with wild-type HRAS and mutant HRAS (Q61L) retroviral constructs and performed western blot assay. We found that overexpression of either wild-type HRAS or mutant HRAS increased phospho-Erk 42/44 and decreased both Casz1 mRNA and protein levels (revised Fig. 2c). This demonstrates that over-expression of wild-type or mutant RAS in C2C12 cells activates the RAS-MEK pathway, which leads to the repression of Casz1 expression.

Pages 7-9, Figs. 2d-h and 3a-e: It is not clear how the genome-wide binding of CASZ1 is determined. Since CASZ1 is shown to play a role in differentiation systems other than myogenic one (as described in the introduction and discussion sections), there should be tissue/cell type-specific mechanisms, which might involve myogenic transcription factors such as MYOD. The authors should more discuss a potential mechanism on this point: e.g., if CASZ1 interacts with MYOD or other transcription factors, if CASZ1 binding is upstream or downstream of MYOD and/or MYOG binding (Fig. 3b-d for instance

indicates MEKi treatment appears to increase MYOG, but not MYOD, binding), how “a cooperative role of CASZ1 with MYOD, MYOG and MEF2D” (Page 17) is regulated.

The reviewer queries “...how the genome-wide binding of CASZ1 is determined”. The genome-wide binding of CASZ1 is determined by performing ChIP-seq using anti-CASZ1 antibody in SMS-CTR cells that treated with MEK inhibitor. We revised the legend to Fig. 2d to clarify this.

In response to the reviewers request to discuss a potential mechanism as to whether CASZ1 physically interacts with MYOD, we performed a pull-down (immunoprecipitation) experiment. We performed a co-immunoprecipitation (co-IP) by using anti-CASZ1 antibody in CASZ1b over-expressed SMS-CTR cells. We found that although the IP of CASZ1b could pull down Histone 3, a known protein interactor of CASZ1, there is no “pull-down” of MYOD and MYOG (revised Suppl. Fig. 4g), which indicates that CASZ1 does not physically interact with MYOD or MYOG. Based on our data, we believe that the mechanism of CASZ1 cooperation with MYOD and MYOG is through forming a core regulatory circuitry (CRC). CASZ1, MYOD and MYOG not only bind to their own enhancer region and regulate their own gene expression, but also bind to each other’s transcriptional regulatory elements and regulate each other’s expression (Fig. 1c,j, Fig. 3b-d) to form an interconnected auto-regulatory feed-forward loops (Fig. 3e). Thus, CASZ1, MYOD and MYOG could be each other’s upstream or downstream regulator in an auto-regulatory feed-forward loop. Moreover, as a component of a CRC, these transcription factors work as a group to up-regulate the expression of skeletal muscle genes through binding to the super-enhancers (SEs) of these genes (Suppl. Fig. d). MEF2D is another myogenic regulatory factor, despite we do not have ChIP-seq data to show that MEF2D is a component of this CRC, we did observe that CASZ1 can directly regulate MEF2D expression (Fig. 6f, Suppl. Fig. 6a), suggesting that MEF2D could also contribute to CASZ1-induced skeletal muscle differentiation. We have included these observations in the result section (page 12) and discussion section (page 19-20).

Pages 10-13, Fig. 5: Related to the point above, the molecular roles of CASZ1 on genome are not well documented. CASZ1 not only binds to myogenic genes for the formation of active chromatin environments, but also occupies neural genes to somehow negatively regulate their expression. These opposite actions of CASZ1 are intriguing but may need to be a bit more interpreted or molecularly dissected, although this point is discussed in the Discussion section (Page 16). For instance, the authors could more analyze chromatin accessibility (ATAC-seq) data in Fig. 5 by detecting differentially accessible regions to identify specific genomic loci/genes that are positively or negatively regulated by CASZ1.

Our results show that CASZ1 directly binds to the promoter, enhancer and/or super-enhancer regions of MRFs and skeletal muscle genes, and the binding of CASZ1 on these regions is accompanied with an increased chromatin accessibility and transcriptional activation shown by increased ATAC-seq, H3K27ac, RNA Pol II and RNA-seq signals at these regions (Fig. 5f, Fig. 6e, Suppl. Fig. 6a). Thus, CASZ1 up-regulates skeletal muscle genes through regional epigenetic modifications. The molecular mechanisms of how the binding of CASZ1 on its target genes affect chromatin status remain unclear, but we speculate that CASZ1 functions through recruiting co-activators or co-repressors to open or close the chromatins. Our next major efforts are to answer these questions.

As recommended, we focused on the effect of restoration of CASZ1b in SMS-CTR cells on global chromatin accessibility (ATAC-seq) and gene expression changes (RNA-seq) without considering whether there is a CASZ1b binding signal or not. We did not observe a significant positive correlation between chromatin accessibility and gene transcription regulation when we compared ATAC-seq data and RNA-seq data. We think this might be because at the 48 hr time point we analyzed the chromatin status changes, and the gene expression changes were in flux. In our future studies, we will perform a time course experiment for ATAC-seq and RNA-seq. We described this observation in the discussion section (page 19).

There are some typos throughout the manuscript (e.g., middle of Page 13, “differentiation and proliferation of muscle cells of muscle cells”). Please check the text carefully again.

Thanks for pointing this out. We have corrected these typos.

Reviewer #2 (Remarks to the Author):

In this study, Liu and colleagues explore the functional role of the Casz1 transcription factor in rhabdomyosarcoma (RMS). Since RMS is related to a skeletal myogenic lineage, the authors examine the expression and regulation of Casz1 in C2 myoblasts. They show that Casz1 is highly induced during differentiation and regulated by myogenic transcription factors MyoD and myogenin, the same factors whose functions are impaired in RMS. Gain or loss of Casz1 expression in C2 myoblasts supported that Casz1 functions as a positive regulator of skeletal muscle differentiation, which the authors further showed was mediated through the control of MyoD and myogenin expression and their downstream gene targets. Thus, Casz1 and the myogenic transcription factors operate in a regulatory loop to control myogenesis.

Next, the authors investigated the relevance of Casz1 in RMS. They particularly focus on the ERMS subtype and find that Casz1 was downregulated in RD cells. Treatment of RD cells with a MEK inhibitor promoted differentiation, concomitant with increases in Casz1, which was reversed with expression of activating mutants of KRas.

By ChIP-Seq analysis of Casz1 binding to chromatin, data showed that enrichment of Casz1 predominated in differentiated ERM cells, at gene sites containing E box consensus sequences and overlapping with enrichment peaks identified with active transcription marks. The same E box sites were predominantly occupied by myogenic transcription factors, MyoD and myogenin. These data suggested that inhibition of Ras-MEK signaling induced differentiation in ERMS, was mediated by cooperative function between Casz1 and myogenic transcription factors, similar to how C2C12 differentiation was controlled.

To assess the human tumor relevance, the authors screened RMS tumors and found a small subset containing Casz1 mutations. They analyzed one of these mutants containing an R25C variant, which was shown to cause the cellular mis-localization of Casz1 and reduce transactivation of myogenic genes. In vivo, the injection of an ERMS cell line expressing the Casz1 R25C variant in mice had no effect on tumor growth as opposed to wt Casz1 which suppressed tumor growth.

The authors conclude that similar to MyoD, Casz1 functions in ERMS as a tumor suppressor by acting cooperatively with myogenic transcription factors to promote differentiation of muscle progenitor cells.

Critique

Overall, this is a carefully performed study that provides insight into the function of Casz1 and its relation to RMS and skeletal muscle differentiation. There are no major limitations in the study, but the points below refer to areas in the manuscript that were not sufficiently addressed.

1. It was unclear whether Ras-MEK signaling is sufficient to regulate Casz1 expression. The authors should perform some gain and loss of function experiments in C2C12 myoblasts to address whether Ras-MEK signaling can regulate Casz1.

In the revised Fig. 2c we show that overexpression of HRAS wild-type or mutant construct in C2C12 cells activates RAS-MEK pathway and results in a decrease of Casz1 expression. We also performed a

loss of function study in C2C12 cells to show that either genetic knockdown of Mek1 or pharmaceutical inhibition of Mek using trametinib results in an increase of Casz1 expression (revised Fig. 2b and revised suppl. Fig. 2i). We also treated a KRAS mutated neuroblastoma cell line (KRAS G12D) with the MEK inhibitor trametinib and also observed an up regulation of CASZ1 (revised Suppl. Fig. 2k). These results further confirm that RAS-MEK signaling can regulate Casz1 expression even in different cell types.

2. It was also unclear why the authors solely focused on ERMS. They should check whether Casz1 expression levels are altered in ARMS. Given that MyoD function is impaired in both ERMS and ARMS subtypes, and that the authors showed that MyoD regulates Casz1, one would suspect that Casz1 might also be relevant in the more aggressive form of RMS.

We mainly focused on ERMS because the mRNA levels of CASZ1 are significantly lower in ERMS compared to normal muscle based on the publicly available microarray, RNA-seq and our own realtime-PCR data (Suppl. Fig. 2b-d). However, the mRNA levels of CASZ1 in some of the ARMS tumor samples are relatively lower compared to normal muscle (Suppl. Fig. 2b-d). To investigate whether CASZ1 is relevant in the more aggressive form of the RMS, we generated a tetracycline (Tet) inducible CASZ1b overexpression stable clone in an ARMS cell line (RH30). We found that over-expression of CASZ1b in RH30 cells results in a decrease in cell proliferation and an up-regulation of skeletal muscle differentiation markers MHC, ACTA1 and CKM, as well as the myogenic regulatory factors MYOD and MYOG (revised Suppl. Fig. 4a-c), which indicates that CASZ1 induces muscle differentiation and suppresses tumor growth in both ERMS and ARMS. We included lines in the results section (page 11) to address this.

3. The relevance of the findings would be significantly enhanced if evidence of Casz1 regulation was confirmed in ERMS patient samples (by qRT-PCR or IHC).

We appreciate this recommendation. To evaluate mRNA levels of CASZ1 in primary RMS patient samples, we performed qRT-PCR on three normal skeletal muscle RNA samples, five primary ERMS and five primary ARMS RNA samples. The results showed that mRNA levels of CASZ1 is significantly lower in ERMS compared to normal skeletal muscle (revised Suppl. Fig. 2d). This result is consistent with what we have found in the publicly available microarray data and RNA-seq data (Suppl. Fig. 2b,c).

Minor comments

a. The authors mentioned using an RMS integrated database, but the identity and details of this resource were not described in the methods section.

Thanks for pointing this out. We received this data from Integrated Rhabdomyosarcoma Databases (iRDb), the Childhood Solid Tumor Network at St. Jude, and have detailed this in the main text when we first mentioned this RMS integrated database (page 6-7).

b. Line 209: the results in this sentence are confusing since Casz1 expression would be expected to be lower in ERMS.

We went through the manuscript and were unable to find this concern in Line 209. We searched for “ERMS” and did not see anything we thought reflected this comment by the reviewer. However, we did go through our manuscript to correct any misleading sentences.

Reviewer #3 (Remarks to the Author):

In this manuscript, Liu et al characterized the role of CASZ1, a zinc finger transcription factor, in skeletal muscle and embryonal rhabdomyosarcoma (ERMS) differentiation by genetic (knockdown and overexpression) and integrative genomic studies. They first showed that CASZ1 is required for the upregulation of MYOD signature genes and skeletal muscle differentiation in normal C2C12 myoblast cells and ERMS cell lines. They noted that the expression of oncogenic RAS proteins impaired the expression of CASZ1 in ERMS cells. Then by integrative analysis of changes in gene expression (by RNA-seq), chromatin accessibility (by ATAC-seq), chromatin occupancy of key TFs and histone marks (by ChIP-seq), they provided evidence that CASZ1 directly upregulates skeletal muscle related genes and represses non-muscle genes such as neural lineage related genes. CASZ1 also forms a core regulatory circuitry with MYOD and MYOG in an autoregulatory loop to control enhancer and super-enhancer activity.

Finally, they identified several genetic variants affecting CASZ1 coding sequences in ERMS tumor samples, and provided evidence that one of them (R25C) results in impaired CASZ1 cellular localization as the possible mechanism in RMS tumorigenesis.

Overall, this study presents a set of interesting findings and a nice synthesis of the various epigenomic profiling results in myoblasts and ERMS cells with CASZ1 manipulations. The initial characterization of CASZ1 in normal myoblast differentiation was rigorous and the results were carefully discussed. The analysis of RNA-seq, ATAC-seq and ChIP-seq datasets, and the integrative analysis of multiple datasets were thorough and well executed. The detailed analysis of enhancer and super-enhancer signatures upon CASZ1 depletion or restore expression in ERMS cells provides new insights into the genomic mechanisms by which CASZ1 regulates the myogenic differentiation programs in normal myoblasts and ERMS cells. The identification and characterization of genetic variants affecting CASZ1 function in ERMS patients is an important finding that provides a mechanistic explanation for at least one genetic variant R25C that impairs the tumor suppressive function of CASZ1 in ERMS. In this regard, this work represents an important advance in our understanding of this important transcription factor in the context of myogenesis and ERMS tumorigenesis. There are several questions remain, that if addressed to further improve the already developed manuscript, this work may have an excellent impact.

Major points:

1. The authors provide convincing evidence that activated RAS-MEK signaling by overexpression of oncogenic RAS leads to the suppression of CASZ1 expression in ERMS cell lines. Since oncogenic RAS-MEK also inhibits skeletal muscle differentiation, whereas inhibition of RAS-MEK (i.e. by MEKi or MEK1 knockdown) induces skeletal muscle differentiation, it is important to determine whether the suppression of CASZ1 expression by RAS-MEK is due to direct or indirect effect. This is relevant because the expression of CASZ1 is significantly and progressively upregulated during skeletal muscle differentiation, it is likely that the observed effect on CASZ1 expression was due to impaired differentiation rather than direct regulation of CASZ1. Also, is the suppressive effect on CASZ1 by RAS-MEK specific to myoblasts or ERMS cells or is it also seen in other cellular contexts? It would be interesting to determine whether mutant RAS also suppresses CASZ1 expression in cell/tumor types where CASZ1 is also known to be important and RAS mutations are commonly found. This line of investigation may help address the above question whether the RAS-MEK-mediated suppression of CASZ1 is due to direct or indirect mechanisms.

We appreciate the reviewer's insightful comments. We believe that the regulation of CASZ1 by RAS-MEK is not just a side-effect of cell differentiation for the following reason: when we over-express mutant RAS in C2C12 myoblasts and culture the cells in the regular growth medium (GM), which does not affect cell differentiation, we still observed the down-regulation of Casz1 by RAS (revised Fig. 2c and revised Suppl. Fig. 2j, red bars). CASZ1 is a tumor suppressor in neuroblastoma. To investigate whether

the suppressive effect on CASZ1 by RAS-MEK specific to myoblast or ERMS cells, we treated KRAS mutated neuroblastoma cell line NBEB (KRAS G12D) with MEK inhibitor, we observed an up-regulation of CASZ1 expression (revised Suppl. Fig. 2k), which indicates that mutant RAS suppresses CASZ1 expression in other tumor types. Taken together, these results suggest that the RAS-MEK pathway can directly repress CASZ1 expression.

2. Along the same line, when discussing gene expression changes upon CASZ1 modulations (i.e. knockdown, overexpression or restored expression), it would be helpful to further clarify whether the observed changes are likely due to direct effects by CASZ1-mediated transcriptional control or indirect effects due to impaired cell differentiation. Since many of the affected genes or gene signatures are related to cell differentiation, and only a small subset of differentially expressed genes had CASZ1 chromatin binding by ChIP-seq (Fig. 5), it seems that most of the observed gene expression changes are due to indirect effect on myoblast differentiation.

We think CASZ1 both directly and indirectly regulates skeletal muscle differentiation genes to induce cell differentiation. In Fig. 5b we only showed a few examples of skeletal muscle genes directly regulated by CASZ1b in CTR cells. In fact, when we focus on the 98 skeletal muscle differentiation genes that were up-regulated by CASZ1b based on gene set enrichment assay (Fig. 4e,f and revised Suppl. Table 7), around 40 genes have an average of three CASZ1b binding sites based on CASZ1 ChIP-seq (revised Suppl. Table 11). Thus, CASZ1b directly regulates 40% of the up-regulated skeletal muscle differentiation genes. However, CASZ1 may also indirectly regulated skeletal muscle genes since CASZ1 directly up-regulates myogenic regulatory factors MYOD, MYOG and MEF2D. These genes are well-known direct regulators of skeletal muscle differentiation genes. This analysis indicates that CASZ1 both directly and indirectly up-regulates skeletal muscle differentiation genes to induce cell differentiation. We described these observations on page 14.

3. Most of the integrative analysis of RNA-seq, ChIP-seq and ATAC-seq were performed in the context of ERMS cells, although the authors did analyze gene expression changes (by RNA-seq) and epigenetic changes (by ChIP-seq) in C2C12 myoblast cells (Fig. 1 and Supplementary Fig. 5j). It would be helpful to spend some efforts to have more in-depth analysis of CASZ1-mediated transcriptional programs using these datasets, and compare the results with that in ERMS cells. It would be interesting to know whether there are shared or specific programs that are regulated by CASZ1 in normal myoblast differentiation vs ERMS rhabdomyosarcoma differentiation, which will help elucidate the function of CASZ1 in normal vs tumor cells.

We appreciate this interesting advice. The epigenetic studies were done in different conditions for C2C12 cells (loss of Casz1, cultured in differentiation medium) and ERMS cells (overexpression of CASZ1b, cultured in regular growth medium). To compare the shared or specific programs that are regulated by CASZ1 in normal myoblast differentiation vs ERMS rhabdomyosarcoma differentiation, we focused on RNA-seq studies since these experiments were performed under similar condition: overexpression of CASZ1b in C2C12 cells for 48 hr vs. overexpression of CASZ1b in SMS-CTR cells for 48 hr. We analyzed the RNA-seq data through gene set enrichment analysis (GSEA) by focusing on the C2 curated gene sets (including 4762 gene sets) in detail. We found that many shared programs are regulated by CASZ1b in both C2C12 and SMS-CTR cells. For example, CASZ1b-up-regulated genes are enriched in MYOD targets; CASZ1b-repressed genes are enriched in EGFR signaling, cell cycle, E2F targets and RB1 repressed signature genes (revised Suppl. Table 8). There are some genes sets that are differentially regulated by CASZ1b in myoblasts vs. in ERMS. For example, among the top negatively enriched gene sets, we found that in C2C12 cells CASZ1b regulates gene sets involved in repressing oligodendrocyte differentiation, telomere maintenance, meiotic recombination, chromosome maintenance gene sets, but in SMS-CTR cells the top negatively enriched gene sets regulated by CASZ1 are involved in breast cancer development, colorectal cancer development, cervical cancer proliferation (revised Suppl.

Table 8). Taken together, CASZ1 induces a skeletal muscle differentiation program in both myoblasts and ERMS, however, CASZ1 preferably regulates normal development processes in myoblasts and preferentially regulates oncogenic pathways ERMS. We have addressed this on page 12.

Minor points:

1. In Supplementary Fig. 5f, the authors concluded that restoration of CASZ1b in SMS-CTR cells did not significantly affect global ChIP-seq and ATAC-seq signals, whereas in Fig. 5d,e the correlative changes in H3K27ac, RNA Pol II and ATAC-seq were observed at CASZ1b bound and differentially expressed genes. This is somewhat confusion and should be clearly stated that different genes or genomic regions were used for these analyses.

We appreciate the reviewer pointing out this out. In Supplementary Fig. 5f, the heatmap shows all CASZ1b binding peaks and the aligned peaks of H3K27ac, RNA Pol II, ATAC-seq at CASZ1b peak center (15,772 peaks associated with 6271 genes). No significant changes were observed in H3K27ac, RNA Pol II, ATAC-seq signals before (-) and after (+) Dox treatment if focused on all these 15,772 sites. However, in Fig. 5d,e when we only focused on a list of differentially expressed genes (~600) that are associated with CASZ1b binding sites (~1,200 peaks). Here we observed changes in H3K27ac, RNA Pol II, ATAC-seq signals (Fig. 5d,e). To clarify this observation, we labeled the number of CASZ1b binding sites in the heatmap of Supplementary Fig. 5f and labeled the number of CASZ1b binding sites in the composite plot in Fig. 5d,e. We also described this in more details in the main text (page 13-14).

2. It would be helpful to also comment on possible functions of other genetic variants identified in this study. Are they predicted as deleterious mutations? Do they impact specific protein domains and/or conserved residues?

We have added more detailed information of the genetic variants of CASZ1 on page 16-17 as the following: “The rest of the SNVs identified from RMS result in the following amino acid (AA) changes: E323D, G676C and M1129T. Among the four SNVs, R25C, E323D and G676C are predicted as deleterious mutations based on both SIFT and PPH2 assay. *In silico* analysis showed that R25C localizes to a nuclear localization signal (NLS) (AA 24-43) and G676C localizes to the zinc finger 4 C2H2 domain (ZF4) (AA 668-292) (Suppl. Fig. 7a,b), and both of these regions are highly conserved. For example, *xenopus* Casz1 has a ~80% similarity with human CASZ1b whole protein, however, the NLS and ZF4 in *xenopus* Casz1 is identical to human CASZ1.”

REVIEWERS' COMMENTS:

Reviewer #1 (Remarks to the Author):

The authors have well addressed the points raised by me as well as the other reviewers by performing additional experiments and conducting more analyses on NGS data, all of which further support the findings in this study. Overall, I feel the manuscript has been significantly improved and thus is now suitable for acceptance.

Reviewer #2 (Remarks to the Author):

The manuscript by Liu and colleagues is much improved. The authors have included additional data to strengthen their overall conclusion regarding the role of Casz1 in rhabdomyosarcoma. My concerns from the original version have all been addressed. I would just ask the authors to verify the link that was provided on page 7, which did not appear to be active.

<https://www.stjude.org/research/resources-data/childhood-solid-tumor-network/available-resources.html#irdb>)

Reviewer #3 (Remarks to the Author):

The authors are to be commended for the extraordinary efforts in revising this manuscript by performing a large amount of new experiments and analyses to address the comments from the previous review. The new data significantly strengthened the conclusions related to the mechanistic link between RAS-MEK, CASZ1, and MYOD/MYOG in regulating skeletal muscle differentiation and ERMS. The additional analyses also helped to clarify the potential direct and indirect effects in gene expression and epigenetic changes due to CASZ1 modulation. Overall the new data and textual revisions significantly enhanced the strength of the conclusions, improved clarity of this manuscript, and addressed the original questions. This reviewer recommends the publication of the revised manuscript.

Dear referees,

We really appreciate your comments and critiques, which make our manuscript much stronger. Please find our responses under your comments that highlighted in red.

Sincerely,

Carol J. Thiele & Zhihui Liu

Carol J. Thiele, Ph.D; Zhihui Liu, Ph.D
National Cancer Institute, National Institutes of Health
10 Center Drive, Bldg 10, Rm. 1W-3940, MSC-1105
Bethesda, MD. 20892-1105
Phone: 240-838-3849
Fax: 301-451-7052
E-mail: thielec@mail.nih.gov; liuzhihu@mail.nih.gov

REVIEWERS' COMMENTS:

Reviewer #1 (Remarks to the Author):

The authors have well addressed the points raised by me as well as the other reviewers by performing additional experiments and conducting more analyses on NGS data, all of which further support the findings in this study. Overall, I feel the manuscript has been significantly improved and thus is now suitable for acceptance.

Thanks for the comments.

Reviewer #2 (Remarks to the Author):

The manuscript by Liu and colleagues is much improved. The authors have included additional data to strengthen their overall conclusion regarding the role of Casz1 in rhabdomyosarcoma. My concerns from the original version have all been addressed. I would just ask the authors to verify the link that was provided on page 7, which did not appear to be active.

<https://www.stjude.org/research/resources-data/childhood-solid-tumor-network/available-resources.html#irdb>

Thank you for the comments. This link was moved from Results Section to Data Availability Section to save space. We have verified that the link is active.

Reviewer #3 (Remarks to the Author):

The authors are to be commended for the extraordinary efforts in revising this manuscript by performing a large amount of new experiments and analyses to address the comments from the previous review. The new data significantly strengthened the conclusions related to the mechanistic link between RAS-MEK, CASZ1, and MYOD/MYOG in regulating skeletal muscle differentiation and ERMS. The additional analyses also helped to clarify the potential direct and indirect effects in gene expression and epigenetic changes due to CASZ1 modulation. Overall the new data and textual revisions significantly enhanced the

strength of the conclusions, improved clarify of this manuscript, and addressed the original questions. This reviewer recommends the publication of the revised manuscript.

Thanks for the comments.